# MITIGATING DATASET BIAS BY USING PER-SAMPLE GRADIENT

**Sumyeong Ahn**[*]
KAIST AI
sumyeongahn@kaist.ac.kr

**Seongyoon Kim**[*]
KAIST ISysE
curisam@kaist.ac.kr

**Se-young Yun**
KAIST AI
yunseyoung@kaist.ac.kr

## ABSTRACT

The performance of deep neural networks is strongly influenced by the training dataset setup. In particular, when attributes with a strong correlation with the target attribute are present, the trained model can provide unintended prejudgments and show significant inference errors (*i.e., the dataset bias problem*). Various methods have been proposed to mitigate dataset bias, and their emphasis is on weakly correlated samples, called *bias-conflicting samples*. These methods are based on explicit bias labels provided by humans. However, such methods require human costs. Recently, several studies have sought to reduce human intervention by utilizing the output space values of neural networks, such as feature space, logits, loss, or accuracy. However, these output space values may be insufficient for the model to understand the bias attributes well. In this study, we propose a debiasing algorithm leveraging gradient called **P**er-sample **G**radient-based **D**ebiasing (PGD). PGD is comprised of three steps: (1) training a model on uniform batch sampling, (2) setting the importance of each sample in proportion to the norm of the sample gradient, and (3) training the model using importance-batch sampling, whose probability is obtained in step (2). Compared with existing baselines for various datasets, the proposed method showed state-of-the-art accuracy for the classification task. Furthermore, we describe theoretical understandings of how PGD can mitigate dataset bias.

## 1 INTRODUCTION

*Dataset bias* (Torralba & Efros, 2011; Shrestha et al., 2021) is a bad training dataset problem that occurs when unintended easier-to-learn attributes *(i.e., bias attributes)*, having a high correlation with the target attribute, are present (Shah et al., 2020; Ahmed et al., 2020). This is due to the fact that the model can infer outputs by focusing on the bias features, which could lead to testing failures. For example, most "camel" images include a "desert background," and this unintended correlation can provide a false shortcut for answering "camel" on the basis of the "desert." In (Nam et al., 2020; Lee et al., 2021), samples of data that have a strong correlation (like the aforementioned desert/camel) are called "bias-aligned samples," while samples of data that have a weak correlation (like "camel on the grass" images) are termed "bias-conflicting samples."

To reduce the dataset bias, initial studies (Kim et al., 2019; McDuff et al., 2019; Singh et al., 2020; Li & Vasconcelos, 2019) have frequently assumed a case where labels with bias attributes are provided, but these additional labels provided through human effort are expensive. Alternatively, the bias-type, such as "background," is assumed in (Lee et al., 2019; Geirhos et al., 2018; Bahng et al., 2020; Cadene et al., 2019; Clark et al., 2019). However, assuming biased knowledge from humans is still unreasonable, since even humans cannot predict the type of bias that may exist in a large dataset (Schäfer, 2016). Data for deep learning is typically collected by web-crawling without thorough consideration of the dataset bias problem.

---

[*]Equal contribution

Recent studies (Le Bras et al., 2020; Nam et al., 2020; Kim et al., 2021; Lee et al., 2021; Seo et al., 2022; Zhang et al., 2022b) have replaced human intervention with DNN results. They have identified bias-conflicting samples by using empirical metrics for output space (*e.g.,* training loss and accuracy). For example, Nam et al. (2020) suggested a "relative difficulty" based on per-sample training loss and thought that a sample with a high "relative difficulty" was a bias-conflicting sample. Most of the previous research has focused on the output space, such as feature space (penultimate layer output) (Lee et al., 2021; Kim et al., 2021; Bahng et al., 2020; Seo et al., 2022; Zhang et al., 2022b), loss (Nam et al., 2020), and accuracy (Le Bras et al., 2020; Liu et al., 2021). However, this limited output space can impose restrictions on describing the data in detail.

Recently, as an alternative, model parameter space (*e.g.,* gradient (Huang et al., 2021; Killamsetty et al., 2021b; Mirzasoleiman et al., 2020)) has been used to obtain high-performance gains compared to output space approaches for various target tasks. For example, Huang et al. (2021) used gradient-norm to detect out-of-distribution detection samples and showed that the gradient of FC layer $\in \mathbb{R}^{h \times c}$ could capture joint information between feature and softmax output, where $h$ and $c$ are the dimensions of feature and output vector, respectively. Since the gradients of each data point $\in \mathbb{R}^{h \times c}$ constitute high-dimensional information, it is much more informative than the output space, such as logit $\in \mathbb{R}^c$ and feature $\in \mathbb{R}^h$. However, there is no approach to tackle the dataset bias problem using a gradient norm-based metric.

In this paper, we present a resampling method from the perspective of the per-sample gradient norm to mitigate dataset bias. Furthermore, we theoretically justify that the gradient-norm-based resampling method can be an excellent debiasing approach. Our key contributions can be summarized as follows:

- We propose **P**er-sample **G**radient-norm based **D**ebiasing (PGD), a simple and efficient gradient-norm-based debiasing method. PGD is motivated by prior research demonstrating (Mirzasoleiman et al., 2020; Huang et al., 2021; Killamsetty et al., 2021b) that gradient is effective at finding rare samples, and it is also applicable to finding the bias-conflicting samples in the dataset bias problem (See Section 3 and Appendix E).

- PGD outperforms other dataset bias methods on various benchmarks, such as colored MNIST (CM-NIST), multi-bias MNIST (MBMNIST), corrupted CIFAR (CCIFAR), biased action recognition (BAR), biased FFHQ (BFFHQ), CelebA, and CivilComments-WILD. In particular, for the colored MNIST case, the proposed method yielded higher unbiased test accuracies compared with the vanilla and the best methods by $35.94\%$ and $2.32\%$, respectively. (See Section 4)

- We provide theoretical evidence of the superiority of PGD. To this end, we first explain that minimizing the trace of inverse Fisher information is a good objective to mitigate dataset bias. In particular, PGD, resampling based on the gradient norm computed by the biased model, is a possible optimizer for mitigating the dataset bias problem. (See Section 5)

## 2 DATASET BIAS PROBLEM

**Classification model.** We first describe the conventional supervised learning setting. Let us consider the classification problem when a training dataset $\mathcal{D}_n = \{(x_i, y_i)\}_{i=1}^n$, with input image $x_i$ and corresponding label $y_i$, is given. Assuming that there are $c \in \mathbb{N} \setminus \{1\}$ classes, $y_i$ is assigned to the one element in set $C = \{1, ..., c\}$. Note that we focus on a situation where dataset $\mathcal{D}_n$ does not have noisy samples, for example, noisy labels or out-of-distribution samples (e.g., SVHN samples when the task is CIFAR-10). When input $x_i$ is given, $f(y_i|x_i, \theta)$ represents the softmax output of the classifier for label $y_i$. This is derived from the model parameter $\theta \in \mathbb{R}^d$. The cross-entropy (CE) loss $\mathcal{L}_{\text{CE}}$ is frequently used to train the classifier, and it is defined as $\mathcal{L}_{\text{CE}}(x_i, y_i; \theta) = -\log f(y_i|x_i, \theta)$ when the label is one-hot encoded.

**Dataset bias.** Let us suppose that a training set, $\mathcal{D}_n$, is comprised of images, as shown in Figure 1, and that the objective is to classify the `digits`. Each image can be described by a set of attributes, (*e.g.,* for the first image in Figure 1, it can be {`digit 0`, `red`, `thin`,...}). The purpose of the training classifier is to find a model parameter $\theta$ that correctly predicts the target attributes, (*e.g.,* `digit`). Notably, the target attributes are also interpreted as *classes*. However, we focus on a case wherein another attribute that is strongly correlated to the target exists, and we call these attributes *bias attributes*. For example, in Figure 1, the bias attribute is `color`. Furthermore, samples

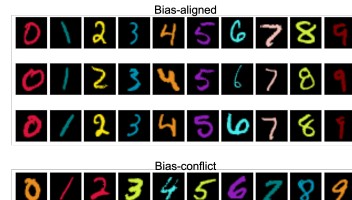

Figure 1: Target and bias attribute: `digit shape`, `color`.

whose bias attributes are highly correlated to the target attributes are called *bias-aligned* (top three rows in Figure 1). Conversely, weakly correlated samples are called *bias-conflicting* (see the bottom row of Figure 1). Therefore, our main scope is that the training dataset which has samples whose bias and target attributes are misaligned. [1] According to (Nam et al., 2020), when the bias attributes are easier-to-learn than the target attributes, dataset bias is problematic, as the trained model may prioritize the bias attributes over the target attributes. For example, for a model trained on the images in Figure 1, it can output class 4 when the (`Orange`, 0) image (*e.g.,* left bottom image) is given, due to the wrong priority, `color` which is an easier-to-learn attribute (Nam et al., 2020).

## 3 PGD: PER-SAMPLE GRADIENT-NORM-BASED DEBIASING

In this section, we propose a novel debiasing algorithm, coined as PGD. PGD consists of two models, biased $f_b$ and debiased $f_d$, with parameters $\theta_b$ and $\theta_d$, respectively. Both models are trained sequentially. Obtaining the ultimately trained debiased model $f_d$ involves three steps: (1) train the biased model, (2) compute the sampling probability of each sample, and (3) train the debiased model. These steps are described in Algorithm 1.

**Step 1: Training the biased model.** In the first step, the biased model is trained on the mini-batches sampled from a uniform distribution $U$, similar to conventional SGD-based training, with data augmentation $\mathcal{A}$. The role of the biased model is twofold:

---

**Algorithm 1** PGD: Per-sample Gradient-norm based Debiasing

1: Input: dataset $\mathcal{D}_n$, learning rate $\eta$, iterations $T_b, T_d$, Batch size $B$, Data augmentation operation $\mathcal{A}(\cdot)$, Initial parameter $\theta_0$, GCE parameter $\alpha$
   /** **STEP 1: Train** $f_b$ **/**
2: **for** $t = 1, 2, \cdots, T_b$ **do**
3:    Construct a mini-batch $\mathcal{B}_t = \{(x_i, y_i)\}_{i=1}^B \sim U$.
4:    Update $\theta_t$ as:
      $\theta_{t-1} - \frac{\eta}{B} \nabla_\theta \sum_{(x,y) \in B_t} \mathcal{L}_{\text{GCE}}(\mathcal{A}(x), y; \theta_{t-1}, \alpha)$
5: **end for**
   /** **STEP 2: Calculate** $h$ **/**
6: Calculate $h(x_i, y_i)$ for all $(x_i, y_i) \in \mathcal{D}_n$, (1).
   /** **STEP 3: Train** $f_d$ **based on** $h$ **/**
7: **for** $t = 1, 2, \cdots, T_d$ **do**
8:    Construct a mini-batch $\mathcal{B}'_t = \{(x_i, y_i)\}_{i=1}^B \sim h$.
9:    Update $\theta_{T_b+t}$ as:
      $\theta_{T_b+t-1} - \frac{\eta}{B} \nabla_\theta \sum_{(x,y) \in \mathcal{B}'_t} \mathcal{L}_{\text{CE}}(\mathcal{A}(x), y; \theta_{T_b+t-1})$
10: **end for**

---

it detects which samples are bias-conflicting and calculates how much they should be highlighted. Therefore, we should make the biased model uncertain when it faces bias-conflicting samples. In doing so, the biased model, $f_b$, is trained on the generalized cross-entropy (GCE) loss $\mathcal{L}_{\text{GCE}}$ to amplify the bias of the biased model, motivated by (Nam et al., 2020). For an input image $x$, the corresponding true class $y$, $\mathcal{L}_{\text{GCE}}$ is defined as $\mathcal{L}_{\text{GCE}}(x, y; \theta, \alpha) = \frac{1 - f(y|x, \theta)^\alpha}{\alpha}$. Note that $\alpha \in (0, 1]$ is a hyperparameter that controls the degree of emphasizing the easy-to-learn samples, namely bias-aligned samples. Note that when $\alpha \to 0$, the GCE loss $\mathcal{L}_{\text{GCE}}$ is exactly the same as the conventional CE loss $\mathcal{L}_{\text{CE}}$. We set $\alpha = 0.7$ as done by the authors of Zhang & Sabuncu (2018), Nam et al. (2020) and Lee et al. (2021).

**Step 2: Compute the gradient-based sampling probability.** In the second step, the sampling probability of each sample is computed from the trained biased model. Since rare samples have large gradient norms compared to the usual samples at the biased model (Hsu et al., 2020), the sampling probability of each sample is computed to be proportional to its gradient norm so that bias-conflicting samples are over-sampled. Before computing the sampling probability, the per-sample gradient with respect to $\mathcal{L}_{\text{CE}}$ for all $(x_i, y_i) \in \mathcal{D}_n$ is obtained from the biased model. We propose the following sampling probability of each sample $h(x_i, y_i)$ which is proportional to their gradient norms, as follows:

$$h(x_i, y_i) = \frac{\|\nabla_\theta \mathcal{L}_{\text{CE}}(x_i, y_i; \theta_b)\|_s^r}{\sum_{(x_i, y_i) \in \mathcal{D}_n} \|\nabla_\theta \mathcal{L}_{\text{CE}}(x_i, y_i; \theta_b)\|_s^r}, \tag{1}$$

where $\|\cdot\|_s^r$ denotes $r$ square of the $L_s$ norm, and $\theta_b$ is the result of Step 1. Since, $h(x_i, y_i)$ is the sampling probability on $\mathcal{D}_n$, $h(x_i, y_i)$ is the normalized gradient-norm. Note that computing the gradient for all samples requires huge computing resources and memory. Therefore, we only extract the gradient of the final FC layer parameters. This is a frequently used technique for reducing the computational complexity (Ash et al., 2019; Mirzasoleiman et al., 2020; Killamsetty et al., 2021b;a; 2020). In other words, instead of $h(x_i, y_i)$, we empirically utilize $\hat{h}(x_i, y_i) = \frac{\|\nabla_{\theta_{\text{fc}}} \mathcal{L}_{\text{CE}}(x_i, y_i; \theta_b)\|_s^r}{\sum_{(x_i, y_i) \in \mathcal{D}_n} \|\nabla_{\theta_{\text{fc}}} \mathcal{L}_{\text{CE}}(x_i, y_i; \theta_b)\|_s^r}$, where $\theta_{\text{fc}}$ is the parameters of the final FC layer. We consider $r = 1$ and $s = 2$ (*i.e.,* $L_2$), and deliver ablation studies on various $r$ and $s$ in Section 4.

**Step 3: Ultimate debiased model training.** Finally, the debiased model, $f_d$, is trained using mini-batches sampled with the probability $h(x_i, y_i)$ obtained in Step 2. Note that, as described in

---

[1]Note that bias-alignment cannot always be strictly divisible in practice. For ease of explanation, we use the notations bias-conflicting/bias-aligned.

Algorithm 1, our debiased model inherits the model parameters of the biased model $\theta_{T_b}$. However, Lee et al. (2021) argued that just oversampling bias-conflicting samples does not successfully debias, and this unsatisfactory result stems from the data diversity (*i.e.,* data augmentation techniques are required). Hence, we used simple randomized augmentation operations $\mathcal{A}$ such as random rotation and random color jitter to oversample the bias-conflicting samples.

# 4    EXPERIMENTS

In this section, we demonstrate the effectiveness of PGD for multiple benchmarks compared with previous proposed baselines. Detail analysis not in this section (*e.g.,* training time, unbiased case study, easier to learn target attribute, sampling probability analysis, reweighting with PGD) are described in the Appendix E.

## 4.1    BENCHMARKS

To precisely examine the debiasing performance of PGD, we used the Colored MNIST, Multi-bias MNIST, and Corrupted CIFAR datasets as synthetic datasets, which assume situations in which the model learns bias attributes first. BFFHQ, BAR, CelebA, and CivilComments-WILDS datasets obtained from the real-world are used to observe the situations in which general algorithms have poor performance due to bias attributes. Note that BFFHQ and BAR are biased by using human prior knowledge, while CelebA and CivilComments-WILDS are naturally biased datasets. A detailed explanation of each benchmark are presented in Appendix A.

**Colored MNIST (CMNIST).** CMNIST is a modified version of MNIST dataset (LeCun et al., 2010), where `color` is the biased attribute and `digit` serves as the target. We randomly selected ten colors that will be injected into the digit. Evaluation was conducted for various ratios $\rho \in \{0.5\%, 1\%, 5\%\}$, where $\rho$ denotes the portion of bias-conflicting samples. Note that CMNIST has only one bias attribute: `color`.

**Multi-bias MNIST (MB-MNIST).** The authors of (Shrestha et al., 2021) stated that CMNIST is too simple to examine the applicability of debiaising algorithms for complex bias cases. However, the dataset that Shrestha et al. (2021) generated is also not complex, since they did not use an real-world pattern dataset (*e.g.,* MNIST) and used simple artificial patterns (*e.g.,* straight line and triangle). Therefore, we generated a MB-MNIST; we used benchmark to reflect the real-worled better than (Shrestha et al., 2021). MB-MNIST consists of eight attributes: digit (LeCun et al., 2010), alphabet (Cohen et al., 2017), fashion object (Xiao et al., 2017), Japanese character (Clanuwat et al., 2018), digit color, alphabet color, fashion object color, Japanese character color. Among the eight attributes, the target attribute is digit shape and the others are the bias attributes. To construct MB-MNIST, we follow the CMNIST protocol, which generates bias by aligning two different attributes (*i.e.,* digit and color) with probability $(1 - \rho)$. MB-MNIST dataset is made by independently aligning the digit and seven other attributes with probabity $(1 - \rho)$. Note that rarest sample is generated with probability $\rho^7$. When $\rho$ is set as the CMNIST case, it is too low to generate sufficient misaligned samples. Therefore, we use $\rho \in \{10\%, 20\%, 30\%\}$ to ensure the trainability.

**Corrupted CIFAR (CCIFAR).** CIFAR10 (Krizhevsky et al., 2009) is comprised of ten different objects, such as an `airplane` and a `car`. Corrupted CIFAR are biased with ten different types of texture bias (*e.g.,* frost and brightness). The dataset was constructed by following the design protocol of (Hendrycks & Dietterich, 2019), and the ratios $\rho \in \{0.5\%, 1\%, 5\%\}$ are used.

**Biased action recognition (BAR).** Biased action recognition dataset was derived from (Nam et al., 2020). It comprised six classes for action, (*e.g.,* `climbing` and `diving`), and each class is biased with place. For example, `diving` class pictures are usually taken `underwater`, while a few images are taken from the `diving pool`.

**Biased FFHQ (BFFHQ).** BFFHQ dataset was constructed from the facial dataset, FFHQ (Karras et al., 2019). It was first proposed in (Kim et al., 2021) and was used in (Lee et al., 2021). It is comprised of two gender classes, and each class is biased with age. For example, most female pictures are *young* while male pictures are *old*. This benchmark follows $\rho = 0.5\%$.

**CelebA.** CelebA (Liu et al., 2015) is a common real-world face classification dataset. The goal is classifying the hair color ("blond" and "not blond") of celebrities which has a spurious correlation with the gender ("male" or "female") attribute. Hair color of almost all female images is blond. We report the average accuracy and the worst-group accuracy on the test dataset.

Table 1: Average test accuracy and standard deviation (three runs) for experiments with the MNIST variants under various bias conflict ratios. The best accuracy is indicated in **bold** for each case.

| Dataset | $\rho$ | Vanilla | LfF | JTT | Disen | PGD (Ours) |
|---|---|---|---|---|---|---|
| CMNIST | 0.5% | $60.94_{\pm 0.97}$ | $91.35_{\pm 1.83}$ | $85.84_{\pm 1.32}$ | $94.56_{\pm 0.57}$ | $\mathbf{96.88_{\pm 0.28}}$ |
| | 1% | $79.13_{\pm 0.73}$ | $96.88_{\pm 0.20}$ | $95.07_{\pm 3.42}$ | $96.87_{\pm 0.64}$ | $\mathbf{98.35_{\pm 0.12}}$ |
| | 5% | $95.12_{\pm 0.24}$ | $98.18_{\pm 0.05}$ | $96.56_{\pm 1.23}$ | $98.35_{\pm 0.20}$ | $\mathbf{98.62_{\pm 0.14}}$ |
| MBMNIST | 10% | $25.23_{\pm 1.16}$ | $19.18_{\pm 4.45}$ | $25.34_{\pm 1.45}$ | $25.75_{\pm 5.38}$ | $\mathbf{61.38_{\pm 4.41}}$ |
| | 20% | $62.06_{\pm 2.45}$ | $65.72_{\pm 6.23}$ | $68.02_{\pm 3.23}$ | $61.62_{\pm 2.60}$ | $\mathbf{89.09_{\pm 0.97}}$ |
| | 30% | $87.61_{\pm 1.60}$ | $89.89_{\pm 1.76}$ | $85.44_{\pm 3.44}$ | $88.36_{\pm 2.06}$ | $\mathbf{90.76_{\pm 1.84}}$ |
| CCIFAR | 0.5% | $23.06_{\pm 1.25}$ | $28.83_{\pm 1.30}$ | $25.34_{\pm 1.00}$ | $29.96_{\pm 0.71}$ | $\mathbf{30.15_{\pm 1.22}}$ |
| | 1% | $25.94_{\pm 0.54}$ | $33.33_{\pm 0.15}$ | $33.62_{\pm 1.05}$ | $36.35_{\pm 1.69}$ | $\mathbf{42.02_{\pm 0.73}}$ |
| | 5% | $39.31_{\pm 0.66}$ | $50.24_{\pm 1.41}$ | $45.13_{\pm 3.11}$ | $51.19_{\pm 1.38}$ | $\mathbf{52.43_{\pm 0.14}}$ |

Table 2: Average test accuracy and standard deviation (three runs) for experiments with the raw image benchmarks: BAR and BFFHQ. The best accuracy is indicated in **bold**, and for the overlapped best performance case is indicated in Underline.

| Dataset | Vanilla | LfF | JTT | Disen | PGD (Ours) |
|---|---|---|---|---|---|
| BAR | $63.15_{\pm 1.06}$ | $64.41_{\pm 1.30}$ | $63.62_{\pm 1.33}$ | $64.70_{\pm 2.06}$ | $\mathbf{65.39_{\pm 0.47}}$ |
| BFFHQ | $77.77_{\pm 0.45}$ | $82.13_{\pm 0.38}$ | $77.93_{\pm 2.16}$ | $82.77_{\pm 1.40}$ | $\mathbf{84.20_{\pm 1.15}}$ |

**CivilComments-WILDS.** CivilComments-WILDS (Borkan et al., 2019) is a dataset to classify whether an online comment is toxic or non-toxic. The mentions of certain demographic identities (male, female, White, Black, LGBTQ, Muslim, Christian, and other religion) cause the spurious correlation with the label.

## 4.2 IMPLEMENTATION.

**Baselines.** We select baselines available for the official code from the respective authors among debiasing methods without prior knowledge on the bias. Our baselines comprise six methods on the various tasks: vanilla network, LfF (Nam et al., 2020), JTT (Liu et al., 2021)[2], Disen (Lee et al., 2021), EIIL (Creager et al., 2021) and CNC (Zhang et al., 2022b).

**Implementation details.** We use three types of networks: two types of simple convolutional networks (SimConv-1 and SimConv-2) and ResNet18 (He et al., 2016). Network imeplementation is described in Appendix B. Colored MNIST is trained on SGD optimizer, batch size 128, learning rate 0.02, weight decay 0.001, momentum 0.9, learning rate decay 0.1 every 40 epochs, 100 epochs training, and GCE parameter $\alpha$ 0.7. For Multi-bias MNIST, it also utilizes SGD optimizer, and 32 batch size, learning rate 0.01, weight decay 0.0001, momentum 0.9, learning rate decay 0.1 with decay step 40. It runs 100 epochs with GCE parameter 0.7. For corrupted CIFAR and BFFHQ, it uses ResNet18 as a backbone network, and exactly the same setting presented by Disen (Lee et al., 2021). [3] For CelebA, we follows experimental setting of (Zhang et al., 2022b) which uses ResNet50 as a backbone network. For CivilComments-WILDS, we utilize exactly the same hyperparameters of (Liu et al., 2021) and utilize pretrained BERT. To reduce the computational cost in extracting the per-sample gradients, we use only a fully connected layer, similar to (Ash et al., 2019; Mirzasoleiman et al., 2020; Killamsetty et al., 2021b;a; 2020). Except for CivilComments-WILDS and CelebA, we utilize data augmentation, such as color jitter, random resize crop and random rotation. See Appendix B for more details.

## 4.3 RESULTS AND EMPIRICAL ANALYSIS

**Accuracy results.** In Table 1, we present the comparisons of the image classification accuracy for the unbiased test sets. The proposed method outperforms the baseline methods for all benchmarks and for all different ratios. For example, our model performance is 35.94% better than that of the vanilla model for the colored MNIST benchmark with a ratio $\rho = 0.5\%$. For the same settings, PGD performs better than Disen by 2.32%.

As pointed out in (Shrestha et al., 2021), colored MNIST is too simple to evaluate debiasing performance on the basis of the performance of baselines. In Multi-bias MNIST case, other models fail to

---

[2]In the case of JTT (Liu et al., 2021), although the authors used bias label for validation dataset (especially, bias-conflicting samples), we tune the hyperparameters using a part of the biased training dataset for fair comparison. Considering that JTT does not show significant performance gain in the results, it is consistent with the existing results that the validation dataset is important in JTT, as described in (Idrissi et al., 2022).

[3]Lee et al. (2021) only reported bias-conflicting case for BFFHQ, but we report the unbiased test result.

Table 3: Average and worst test accuracy with the raw image benchmark: **CelebA** and raw NLP task: **CivilComments-WILDS**. The results of comparison algorithms are the results reported in (Zhang et al., 2022b). The best worst accuracy is indicated in **bold**.

| | | Vanilla | LfF | EIIL | JTT | CNC | Ours |
|---|---|---|---|---|---|---|---|
| CelebA | Avg. | 94.9 | 85.1 | 85.7 | 88.1 | 88.9 | 88.6 |
| | Worst | 47.7 | 77.2 | 81.7 | 81.5 | **88.8** | **88.8** |
| CivilComments | Avg. | 92.1 | 92.5 | 90.5 | 91.1 | 81.7 | 92.1 |
| | Worst | 58.6 | 58.8 | 67.0 | 69.3 | 68.9 | **70.6** |

(a) Colored MNIST  (b) Multi-bias MNIST  (c) Corrupted CIFAR

Figure 2: Average PGD results for various of norms, $\{L_1, L_2, L_2^2, L_\infty\}$, for the feature-injected benchmarks. The error bars represent the standard deviation of three independent trials.

obtain higher unbiased test results, even though the ratio is high, *e.g.,* $10\%$. In this complex setting, PGD shows superior performance over other methods. For example, its performance is higher by $36.15\%$ and $35.63\%$ compared with the performance of vanilla model and Disen for the ratio of $10\%$.

Similar to the results for the bias-feature-injected benchmarks, as shown in Table 2 and Table 3, PGD shows competitive performance among all the debiasing algorithms on the raw image benchmark (BAR, BFFHQ, and CelebA). For example, for the BFFHQ benchmark, the accuracy of PGD is $1.43\%$ better than that of Disen. As in Table 3, PGD outperforms the other baselines on CivilComments-WILDs, much more realistic NLP task. Therefore, we believe PGD also works well with transformer, and it is applicable to the real-world.

**Unbiased test accuracy on various norms.** Since, gradient norm can have various candidates, such as order of the norm, we report four configurations of gradient norms. As shown in Figure 2, all norms have significant unbiased test performance. Amongst them, the $L_2$-norm square case shows lower unbiased performance than the other cases. Therefore, it is recommended that any first power of $L_{\{1,2,\infty\}}$-norms be used in PGD for overcoming the dataset bias problem. This is quite different from the results in (Huang et al., 2021), which suggested that $L_1$-norm is the best choice in the research field of out-of-distribution detection.

**Ablation study.** Table 4 shows the importance of each module in our method: generalized cross entropy, and data augmentation modules. We set the ratio $\rho$ as $0.5\%$ and $10\%$ for CMNIST and MB-MNIST, respectively. We observe that GCE is more important that data augmentation for CMNIST. However, data augmentation shows better performance than GCE for MB-MNIST. In all cases, the case where both are utilized outperforms the other cases.

Table 4: Ablation studies on GCE and data augmentation (✓ for applied case).

| GCE | Aug. | CMNIST (0.5%) | MB-MNIST (10%) |
|---|---|---|---|
| | | $84.93_{\pm 0.79}$ | $40.58_{\pm 3.39}$ |
| ✓ | | $93.18_{\pm 1.07}$ | $45.70_{\pm 2.91}$ |
| | ✓ | $91.19_{\pm 0.97}$ | $46.70_{\pm 1.10}$ |
| ✓ | ✓ | $\mathbf{96.88_{\pm 0.28}}$ | $\mathbf{61.38_{\pm 4.41}}$ |

## 5 MATHEMATICAL UNDERSTANDING OF PGD

This section provides a theoretical analysis of per-sample gradient-norm based debiasing. We first briefly summarize the maximum likelihood estimator (MLE) and Fisher information (FI), which are ingredients of this section. We then interpret the debiasing problem as a min-max problem and deduce that solving it the min-max problem can be phrased as minimizing the trace of the inverse FI. Since handling the trace of the inverse FI is very difficult owing to its inverse computation, we look at a glance by relaxing it into a one-dimensional toy example. In the end, we conclude that the gradient-norm based re-sampling method is an attempt to solve the dataset bias problem.

### 5.1 PRELIMINARY

**Training and test joint distributions.** The general joint distribution $\mathcal{P}(x, y|\theta)$ is assumed to be factored into the parameterized conditional distribution $f(y|x, \theta)$ and the marginal distribution $\mathcal{P}(x)$,

which is independent of the model parameter $\theta$, (*i.e.*, $\mathcal{P}(x, y|\theta) = \mathcal{P}(x)f(y|x,\theta)$). We refer to the model $f(y|x,\theta^\star)$ that produces the exact correct answer, as an oracle model, and to its parameter $\theta^\star$ as the oracle parameter. The training dataset $\mathcal{D}_n$ is sampled from $\{(x_i, y_i)\}_{i=1}^n \sim p(x)f(y|x,\theta^\star)$, where the training and test marginal distributions are denoted by $p(x)$ and $q(x)$, respectively. Here, we assume that both marginal distributions are defined on the marginal distribution space $\mathcal{M} = \{\mathcal{P}(x)|\int_{x \in \mathcal{X}} \mathcal{P}(x)\,dx = 1\}$, where $\mathcal{X}$ means the input data space, *i.e.*, $p(x),\ q(x) \in \mathcal{M}$.

**The space $\mathcal{H}$ of sampling probability $h$.** When the training dataset $\mathcal{D}_n$ is given, we denote the sampling probability as $h(x)$ which is defined on the probability space $\mathcal{H}$[4]:

$$\mathcal{H} = \{h(x)\,|\,\textstyle\sum_{(x_i, y_i) \in \mathcal{D}_n} h(x_i) = 1\,,\ h(x_i) \geq 0 \quad \forall (x_i, y_i) \in \mathcal{D}_n\}. \tag{2}$$

**Maximum likelihood estimator (MLE).** When $h(x)$ is the sampling probability, we define MLE $\hat{\theta}_{h(x),\mathcal{D}_n}$ as follows:
$$\hat{\theta}_{h(x),\mathcal{D}_n} = \arg\min_\theta\ -\textstyle\sum_{(x_i,y_i) \in \mathcal{D}_n} h(x_i) \log f(y_i|x_i,\theta).$$
Note that MLE $\hat{\theta}_{h(x),\mathcal{D}_n}$ is a variable controlled by two factors: (1) a change in the training dataset $\mathcal{D}_n$ and (2) the adjustment of the sampling probability $h(x)$. If $h(x)$ is a uniform distribution $U(x)$, then $\hat{\theta}_{U(x),\mathcal{D}_n}$ is the outcome of empirical risk minimization (ERM).

**Fisher information (FI).** FI, denoted by $I_{\mathcal{P}(x)}(\theta)$, is an information measure of samples from a given distribution $\mathcal{P}(x, y|\theta)$. It is defined as follows:

$$I_{\mathcal{P}(x)}(\theta) = \mathbb{E}_{(x,y) \sim \mathcal{P}(x)f(y|x,\theta)}[\nabla_\theta \log\ f(y|x,\theta)\nabla_\theta^\top\ \log\ f(y|x,\theta)]. \tag{3}$$

FI provides a guideline for understanding the test cross-entropy loss of MLE $\hat{\theta}_{U(x),\mathcal{D}_n}$. When the training set is sampled from $p(x)f(y|x,\theta^\star)$ and the test samples are generated from $q(x)f(y|x,\theta^\star)$, we can understand the test loss of MLE $\hat{\theta}_{U(x),\mathcal{D}_n}$ by using FI as follows.

**Theorem 1.** *Suppose Assumption 1 in Appendix F and Assumption 2 in Appendix G hold, then for sufficiently large $n = |\mathcal{D}_n|$, the following holds with high probability:*

$$\mathbb{E}_{(x,y) \sim q(x)f(y|x,\theta^\star)} \left[ \mathbb{E}_{\mathcal{D}_n \sim p(x)f(y|x,\theta^\star)} \left[ -\log f(y|x,\hat{\theta}_{U(x),\mathcal{D}_n}) \right] \right] \leq \tfrac{1}{2n} \mathrm{Tr}\left[ I_{p(x)}(\hat{\theta}_{U(x),\mathcal{D}_n})^{-1} \right] \mathrm{Tr}\left[ I_{q(x)}(\theta^\star) \right]. \tag{4}$$

Here is the proof sketch. The left-hand side of (4) converges to the Fisher information ratio (FIR) $\mathrm{Tr}\left[ I_{p(x)}(\theta^\star)^{-1} I_{q(x)}(\theta^\star) \right]$-related term. Then, FIR can be decomposed into two trace terms with respect to the training and test marginal distributions $p(x)$ and $q(x)$. Finally, we show that the term $\mathrm{Tr}[I_{p(x)}(\theta^\star)^{-1}]$ which is defined in the oracle model parameter can be replaced with $\mathrm{Tr}[I_{p(x)}(\hat{\theta}_{U(x),\mathcal{D}_n})^{-1}]$. The proof of Theorem 1 is in Appendix D. Note that Theorem 1 means that the upper bound of the test loss of MLE $\hat{\theta}_{U(x),\mathcal{D}_n}$ can be minimized by reducing $\mathrm{Tr}[I_{p(x)}(\hat{\theta}_{U(x),\mathcal{D}_n})^{-1}]$.

**Empirical Fisher information (EFI).** In practice, the exact FI (3) cannot be computed since we do not know the exact data generation distribution $\mathcal{P}(x)f(y|x,\theta)$. For practical reasons, the empirical Fisher information (EFI) is commonly used (Jastrzębski et al., 2017; Chaudhari et al., 2019) to reduce the computational cost of gathering gradients for all possible classes when $x$ is given. In the present study, we used a slightly more generalized EFI that involved the sampling probability $h(x) \in \mathcal{H}$ as follows:

$$\hat{I}_{h(x)}(\theta) = \textstyle\sum_{(x_i,y_i) \in \mathcal{D}_n} h(x_i)\nabla_\theta \log\ f(y_i|x_i,\theta)\nabla_\theta^\top \log\ f(y_i|x_i,\theta). \tag{5}$$

Note that the conventional EFI is the case when $h(x)$ is uniform. EFI provides a guideline for understanding the test cross-entropy loss of MLE $\hat{\theta}_{h(x),\mathcal{D}_n}$.

## 5.2 Understanding dataset bias problem via min-max problem

**Debiasing formulation from the perspective of min-max problem.** We formulate the dataset bias problem as described in Definition 1. (6) is a min-max problem formula, a type of robust optimization. Similar problem formulations for solving the dataset bias problem can be found in (Arjovsky et al., 2019; Bao et al., 2021; Zhang et al., 2022a). However, they assume that the training data is divided into several groups, and the model minimizes the worst inference error of the reweighted group dataset. In contrast, the objective of (6) minimizes the worst-case test loss without explicit data groups where the test distribution can be arbitrary.

---

[4]Note that for simplicity, we abuse the notation $h(x, y)$ used in Section 3 as $h(x)$. This is exactly the same for a given dataset $\mathcal{D}_n$ situation.

**Definition 1.** When the training dataset $\mathcal{D}_n \sim p(x)f(y|x,\theta^\star)$ is given, the debiasing objective is

$$\min_{h(x)\in\mathcal{H}} \max_{q(x)\in\mathcal{M}} \mathbb{E}_{(x,y)\sim q(x)f(y|x,\theta^\star)} \left[ -\log f(y|x,\hat{\theta}_{h(x),\mathcal{D}_n}) \right]. \tag{6}$$

The meaning of Definition 1 is that we have to train the model $\hat{\theta}_{h(x),\mathcal{D}_n}$ so that the loss of the worst case test samples ($\max_{q(x)}$) is minimized by controlling the sampling probability $h(x)$ ($\min_{h(x)}$). Note that since we cannot control the given training dataset $\mathcal{D}_n$ and test marginal distribution $q(x)$, the only controllable term is the sampling probability $h(x)$. Therefore, from Theorem 1 and EFI, we design a practical objective function for the dataset bias problem as follows:

$$\min_{h(x)\in\mathcal{H}} \mathrm{Tr}\left[ \hat{I}_{h(x)}(\hat{\theta}_{h(x),\mathcal{D}_n})^{-1} \right]. \tag{7}$$

### 5.3 Meaning of PGD in terms of (7).

In this section, we present an analysis of PGD with respect to (7). To do so, we try to understand (7), which is difficult to directly solve. It is because computing the trace of the inverse matrix is computationally expensive. Therefore, we intuitively understand (7) in the one-dimensional toy scenario.

**One-dimensional example.** We assume that $\mathcal{D}_n$ comprises sets $M$ and $m$ such that elements in each set share the same loss function. For example, the loss functions of the elements in set $M$ and $m$ are $\frac{1}{2}(\theta+a)^2$ and $\frac{1}{2}(\theta-a)^2$ with a given constant $a$, respectively. We also assume that each sample of $M$ and $m$ has the set dependent probability mass $h_M(x)$ and $h_m(x)$, respectively. With these settings, our objective is to determine $h^\star(x) = \arg\min_{h(x)\in\mathcal{H}} \mathrm{Tr}[\hat{I}_{h(x)}(\hat{\theta}_{h(x),\mathcal{D}_n})^{-1}]$. Thanks to the model's simplicity, we can easily find $h^\star(x)$ in a closed form with respect to the gradient at $\hat{\theta}_{U(x),\mathcal{D}_n}$ for each set, *i.e.,* $g_M(\hat{\theta}_{U(x),\mathcal{D}_n})$ and $g_m(\hat{\theta}_{U(x),\mathcal{D}_n})$.

**Theorem 2.** *Under the above setting, the solution of* $(h^\star_M(x), h^\star_m(x)) = \arg\min_{h(x)\in\mathcal{H}} \mathrm{Tr}[\hat{I}_{h(x)}(\hat{\theta}_{h(x),\mathcal{D}_n})^{-1}]$ *is:*

$$h^\star_M(x) = |g_M(\hat{\theta}_{U(x),\mathcal{D}_n})|/Z, \quad h^\star_m(x) = |g_m(\hat{\theta}_{U(x),\mathcal{D}_n})|/Z,$$

*where* $Z = |M||g_M(\hat{\theta}_{U(x),\mathcal{D}_n})| + |m||g_m(\hat{\theta}_{U(x),\mathcal{D}_n})|$, *and* $|M|$ *and* $|m|$ *denote the cardinality of* $M$ *and* $m$, *respectively.*

The proof of Theorem 2 is provided in Appendix E. Note that $h^\star_M(x)$ and $h^\star_m(x)$ are computed using the trained biased model with batches sampled from the uniform distribution $U(x)$. It is the same with the second step of PGD.

**PGD tries to minimize (7).** Theorem 2 implies that (7) can be minimized by sampling in proportion to their gradient norm. Because the basis of PGD is oversampling based on the gradient norm from the biased model, we can deduce that PGD strives to satisfy (7). Furthermore, we empirically show that PGD reduces the trace of the inverse of EFI in the high-dimensional case, as evident in Figure 3.

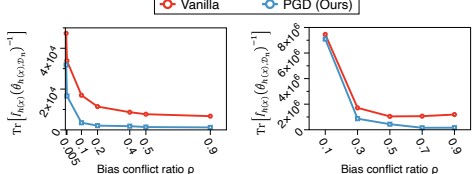

(a) Colored MNIST  (b) Multi-bias MNIST

Figure 3: Target objective $\mathrm{Tr}[\hat{I}_h(\hat{\theta}_{h(x),\mathcal{D}_n})^{-1}]$. PGD : $h(x) = \hat{h}(x)$, and vanilla: $h(x) = U(x)$.

## 6 Related Work

**Debiasing with bias label.** In (Goyal et al., 2017; 2020), a debiased dataset was generated using human labor. Various studies (Alvi et al., 2018; Kim et al., 2019; McDuff et al., 2019; Singh et al., 2020; Teney et al., 2021) have attempted to reduce dataset bias using *explicit bias labels*. Some of these studies (Alvi et al., 2018; Kim et al., 2019; McDuff et al., 2019; Singh et al., 2020; Li et al., 2018; Li & Vasconcelos, 2019), used bias labels for each sample to reduce the influence of the bias labels when classifying target labels. Furthermore, Tartaglione et al. (2021) proposed the EnD regularizer, which entangles target correlated features and disentangles biased attributes. Several studies (Alvi et al., 2018; Kim et al., 2019; Teney et al., 2021) have designed DNNs as a shared feature extractors and multiple classifiers. In contrast to the shared feature extractor methods, McDuff et al. (2019) and Ramaswamy et al. (2021) fabricated a classifier and conditional generative adversarial networks, yielding test samples to determine whether the classifier was biased. Furthermore, Singh

et al. (2020) proposed a new overlap loss defined by a class activation map (CAM). The overlap loss reduces the overlapping parts of the CAM outputs of the two bias labels and target labels. The authors of (Li & Vasconcelos, 2019; Li et al., 2018) employed bias labels to detect bias-conflicting samples and to oversample them to debias. In (Liu et al., 2021), a reconstructing method based on the sample accuracy was proposed. Liu et al. (2021) used bias labels in the validation dataset to tune the hyper-parameters. On the other hand, there has been a focus on fairness within each attribute (Hardt et al., 2016; Woodworth et al., 2017; Pleiss et al., 2017; Agarwal et al., 2018). Their goal is to prevent bias attributes from affecting the final decision of the trained model.

**Debiasing with bias context.** In contrast to studies assuming the explicit bias labels, a few studies (Geirhos et al., 2018; Wang et al., 2018; Lee et al., 2019; Bahng et al., 2020; Cadene et al., 2019; Clark et al., 2019) assumed that the bias context is known. In (Geirhos et al., 2018; Wang et al., 2018; Lee et al., 2019), debiasing was performed by directly modifying known context bias. In particular, the authors of (Geirhos et al., 2018) empirically showed that CNNs trained on ImageNet (Deng et al., 2009) were biased towards the image texture, and they generated stylized ImageNet to mitigate the texture bias, while Lee et al. (2019) and Wang et al. (2018) inserted a filter in front of the models so that the influence of the backgrounds and colors of the images could be removed. Meanwhile, some studies (Bahng et al., 2020; Clark et al., 2019; Cadene et al., 2019), mitigated bias by reweighting bias-conflicting samples: Bahng et al. (2020) used specific types of CNNs, such as BagNet (Brendel & Bethge, 2018), to capture the texture bias, and the bias was reduced using the Hilbert-Schmidt independence criterion (HSIC). In the visual question answering (VQA) task, Clark et al. (2019) and Cadene et al. (2019) conducted debiasing using the entropy regularizer or sigmoid output of the biased model trained on the fact that the biased model was biased toward the question.

**Debiasing without human supervision.** Owing to the impractical assumption that bias information is given, recent studies have aimed to mitigate bias without human supervision (Le Bras et al., 2020; Nam et al., 2020; Darlow et al., 2020; Kim et al., 2021; Lee et al., 2021). Le Bras et al. (2020) identified bias-conflicting samples by sorting the average accuracy of multiple train-test iterations and performed debiasing by training on the samples with low average accuracy. In (Ahmed et al., 2020), each class is divided into two clusters based on IRMv1 penalty (Arjovsky et al., 2019) using the trained biased model, and the deibased model is trained so that the output of two clusters become similar. Furthermore, Kim et al. (2021) used Swap Auto-Encoder (Park et al., 2020) to generate bias-conflicting samples, and Darlow et al. (2020) proposed the modification of the latent representation to generate bias-conflicting samples by using an auto-encoder. Lee et al. (2021) and Nam et al. (2020) proposed a debiasing algorithm weighted training by using a relative difficulty score, which is measured by the per-sample training loss. Specifically, Lee et al. (2021) used feature-mixing techniques to enrich the dataset feature information. Seo et al. (2022) and Sohoni et al. (2020) proposed unsupervised clustering based debiasing method. Recently, contrastive learning based method (Zhang et al., 2022b) and self-supervised learning method (Kim et al., 2022) are proposed. On the other hand, there have been studies (Li & Xu, 2021; Lang et al., 2021; Krishnakumar et al., 2021) that identify the bias attribute of the training dataset without human supervision.

## 7 Conclusion

We propose a gradient-norm-based dataset oversampling method for mitigating the dataset bias problem. The main intuition of this work is that gradients contain abundant information about each sample. Since the bias-conflicting samples are relatively more difficult-to-learn than bias-aligned samples, the bias-conflicting samples have a higher gradient norm compared with the others. Through various experiments and ablation studies, we demonstrate the effectiveness of our gradient-norm-based oversampling method, called PGD. Furthermore, we formulate the dataset bias problem as a min-max problem, and show theoretically that it can be relaxed by minimizing the trace of the inverse Fisher information. We provide empirical and theoretical evidence that PGD tries to solve the problem of minimizing the trace of the inverse Fisher information problem. Despite this successful outcome and analysis, we are still working on two future projects: release approximations, such as a toy example, for understanding PGD and cases where the given training dataset is corrupted, such as with noisy labels. We hope that this study will help improve understanding of researchers about the dataset bias problem.

## ACKNOWLEDGEMENT

This work was supported by Institute of Information & communications Technology Planning & Evaluation (IITP) grant funded by the Korea government(MSIT) [No.2019-0-00075, Artificial Intelligence Graduate School Program(KAIST), 10%] and Institute of Information & communications Technology Planning & Evaluation (IITP) grant funded by the Korea government(MSIT) [No.2022-0-00641, XVoice: Multi-Modal Voice Meta Learning, 90%]

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

# – Appendix –
# Mitigating Dataset Bias by Using Per-sample Gradient

Due to the page constraint, this extra material includes additional results and theoretical proofs that are not in the original manuscript. Section A demonstrates how to create datasets. Section B.1 contains implementation details such as hyperparameters, computer resources, and a brief explanation of the baseline. Section C and Section D include case studies and empirical evidence of PGD. Section E demonstrates additional experiment results. In Section F, we first provide a notation summary and some assumptions for theoretical analysis. Section G and Section H include proofs of Theorem 1 and Theorem 2 with lemmas, respectively.

## A  BENCHMARKS AND BASELINES

We explain the datasets utilized in Section 4. In short, we build MNIST variants from scratch, while others get them directly from online repositories BAR[5], CCIFAR and BFFHQ[6].

### A.1  CONTROLLED BIASED BENCHMARKS

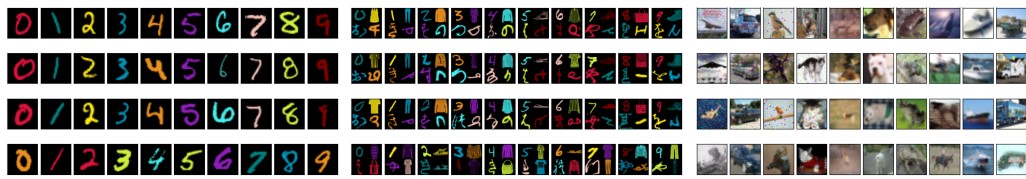

Figure 4: Colored MNIST: The single bias attribute is color, and the target attribute is shape. The top 3 rows represent bias-aligned samples, and the bottom row samples are bias-conflicting examples.

Figure 5: Multi-bias MNIST: Multiple colors and objects bias, with digit shape as the target. The top 3 rows represent bias-aligned samples, and the bottom row samples are bias-conflicting examples.

Figure 6: Corrupted CIFAR: corruption is the bias attribute, while target attribute is object. The top three rows are bias-aligned samples, while the bottom row are bias-conflicting examples.

**Colored MNIST (CMNIST)**  The MNIST dataset (LeCun et al., 2010) is composed of 1-dimensional grayscale handwritten images. The size of the image is $28 \times 28$. We inject color into these gray images to give them two main attributes: color and digit shape. This benchmark comes from related works (Nam et al., 2020; Kim et al., 2021; Lee et al., 2021; Bahng et al., 2020). At the beginning of the generation, ten uniformly sampled RGB colors are chosen, $\{C_i\}_{i \in [10]} \in \mathbb{R}^{3 \times 10}$. When the constant $\rho$, a ratio of bias-conflicting samples, is given, each sample $(x, y)$ is colored by the following steps: (1) Choose bias-conflicting or bias-aligned samples: take a random sample and set it to bias-conflicting set when $u < \rho$ where $u \sim \mathcal{U}(0, 1)$, otherwise bias-aligned. In experiments, we use $\rho \in \{0.5\%, 1\%, 5\%\}$. (2) Coloring: Note that each $C_i \in \mathbb{R}^3$ ($i \in [10]$) is a bias-aligned three-dimensional color vector for each digit $i \in \{10\}$. Then for bias-aligned images with the arbitrary digit $y$, color the digit with $c \sim \mathcal{N}(C_y, \sigma I_{3 \times 3})$. In the case of bias conflicting images with the arbitrary digit $y$, first uniformly sample $C_{U_y} \in \{C_i\}_{i \in [10] \backslash y}$, and color the digit with $c \sim \mathcal{N}(C_{U_y}, \sigma I_{3 \times 3})$. In the experiments, we set $\sigma$ as 0.0001. We use $55,000$ samples for training $5,000$ samples for validation (*i.e.,* $10\%$), and $10,000$ samples for testing. Take note that test samples are unbiased, which means $\rho = 90\%$.

**Multi-Bias MNIST**  Multi-bias MNIST has images with size $56 \times 56$. This dataset aims to test the case where there are multiple bias attributes. To accomplish this, we inject a total of seven bias

---

[5]https://github.com/alinlab/BAR
[6]https://github.com/kakaoenterprise/Learning-Debiased-Disentangled

attributes: digit color, fashion object, fashion color, Japanese character, Japanese character color, English character, and English character color, with digit shape serving as the target attribute. We inject each bias independently into each sample, as with the CMNIST case (*i.e.,* sampling and injecting bias). We also set $\rho = 90\%$ for all bias attributes to generate an unbiased test set. As with CMNIST, we use $55,000$ samples for training and $5,000$ samples for validation, and $10,000$ samples for testing.

**Corrupted CIFAR** This dataset was generated by injecting filters into the CIFAR10 dataset (Krizhevsky et al., 2009). The work (Nam et al., 2020; Lee et al., 2021) inspired this benchmark. In this benchmark, the target attribute and the bias attribute are object and corruption, respectively. {Snow, Frost, Fog, Brightness, Contrast, Spatter, Elastic, JPEG, Pixelate, Saturate} are examples of corruption. We downloaded this benchmark from the repository of the official code of Disen (Lee et al., 2021). This dataset contains $45,000$ in training samples, $5,000$ in validation samples, and $10,000$ in testing images. As with prior datasets, the test dataset is composed of unbiased samples (*i.e.,* $\rho = 90\%$).

## A.2 REAL-WORLD BENCHMARKS

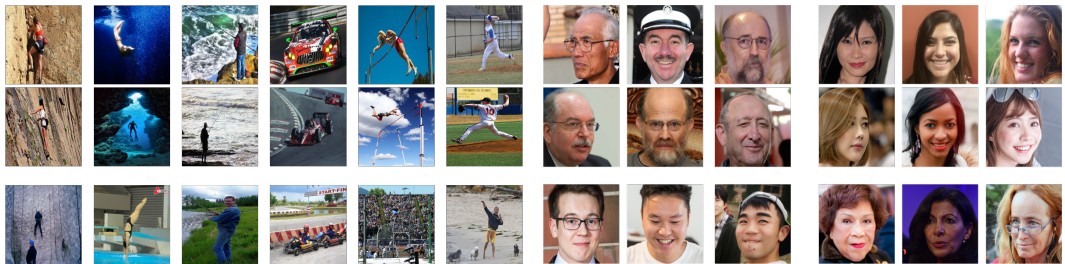

Figure 7: Biased Action Recognition: The biased attribute is background, while the target attribute is action. The top 2 rows are bias-aligned samples, and the bottom row is bias-conflict samples.

Figure 8: BFFHQ: stands for biased FFHQ. Target attribute: gender, biased attribute: age. The samples in the top two rows are bias-aligned, while the samples in the bottom row are bias-conflict.

**Biased Action Recognition (BAR)** This dataset comes from the paper (Nam et al., 2020) for real-world image testing. TThis benchmark aims to classify six actions {Climbing, Diving, Fishing, Racing, Throwing, Vaulting} even though the places are biased. Target and bias attribute pairs are (Climbing,RockWall), (Diving,Underwater), (Fishing, WaterSurface), (Racing, APavedTrack), (Throwing, PlayingField), and (Vaulting, Sky). Bias-conflicting samples, for example, are (Climbing, IceCliff), (Diving, Indoor), (Fishing, Forest), (Racing, OnIce), (Throwing, Cave), (Vaulting, Beach). There are $1,941$ samples for training and $6,54$ samples for testing. To split the training and validation samples, we used $10\%$ validation samples, i.e., $1,746$ images for training and $195$ validation. We download training datasets from the online repository of BAR.

**Biased FFHQ** This BFFHQ benchmark was conducted in (Lee et al., 2021; Kim et al., 2021). Target and bias attributes for bias-aligned samples are (Female, Young), and (Male, Old). Here, "Young" refers to people aged 10 to 29, while "old" refers to people aged 40 to 59. The bias-conflicting samples are (Female, Old) and (Male, Young). The number of training, validation, and test samples are $19,200$, $1,000$, and $1,000$, respectively.

**CelebA** CelebA (Liu et al., 2015) is a common real-world face classification dataset, and each image has $40$ attributes. The goal is to classify the hair color ("blond" and "not blond") of celebrities, which has a spurious correlation with the gender ("male" or "female") attribute. In fact, only $6\%$ of blond hair color images are male. Therefore, ERM shows poor performance on the bias-conflicting samples. We report the average accuracy and the worst-group accuracy on the test dataset.

**CivilComments-WILDS** CivilComments-WILDS (Borkan et al., 2019) is a dataset to classify whether an online comment is toxic or non-toxic. Each sentence is a real online comment, curated on the Civil Comments platform, a comment plug-in for independent news sites. The mentions of certain demographic identities (male, female, White, Black, LGBTQ, Muslim, Christian, and other religion)

cause the spurious correlation with the label. Table 5 indicates the portion of toxic comments for each demographic identity.

| Identity | Male | Female | White | Black | LGBTQ | Muslim | Christian | Other religions |
|---|---|---|---|---|---|---|---|---|
| Portion(%) of toxic | 14.9 | 13.7 | 28.0 | 31.4 | 26.9 | 22.4 | 9.1 | 15.3 |

Table 5: For each demographic identity, the portion of toxic comments in the CivilComments-WILDS.

### A.3 BASELINES

In this section, we briefly describe how the baselines, such as LfF, JTT, Disen, GEORGE, BPA, CNC, and EIIL. Please refer to each paper for a detailed explanation because we briefly explain the algorithms.

(1) LfF (Nam et al., 2020) trains the debiased model by weighting the bias-conflicting samples based on the "relative difficulty", computed by the two loss values from the biased model and the debiased model. To amplify the bias-conflicting samples, the authors employ generalized cross-entropy loss with parameter $\alpha = 0.7$. We implement the LfF algorithm following the code officially offered by the authors. The loss functions that this work proposes are as follows:

$$\mathcal{L}_{\text{LfF}} = W(z)\mathcal{L}_{\text{CE}}(C_d(z, y) + \lambda\mathcal{L}_{\text{GCE}}(C_b(z, y)),$$
$$W(z) = \frac{\mathcal{L}_{\text{CE}}(C_b(z), y)}{\mathcal{L}_{\text{CE}}(C_b(z), y) + \mathcal{L}_{\text{CE}}(C_d(z), y)}.$$

Note that $W(z)$ is a relative difficulty and that GCE is a generalized cross-entropy. $z$ denotes feature, which is the output of the penultimate layer, and $C.$ is a fully connected layer.

(2) JTT (Liu et al., 2021) aims to debias by splitting the dataset into correctly and incorrectly learned samples. To do so, JTT trains the biased model first and splits the given training dataset as follows:

$$\mathcal{D}_{\text{error-set}} = \{(x, y) \text{ s.t. } y_{\text{given}} \neq \arg\max_c f_b(x)[c]\}, \tag{1}$$

The ultimate debiased model is then trained by oversampling $\mathcal{D}_{\text{error-set}}$ with $\lambda_{up}$ times. We set $\lambda_{up}$ for all experiments as $1/\rho$. We reproduce the results by utilizing the official code offered by the authors. The main strength of PGD compared to JTT is that PGD does not need to set a hyperparameter $\lambda_{up}$.

(3) Disen (Lee et al., 2021) aims to debias by generating abundant features from mixing features between samples. To do so, the author trains the biased and debiased model by aggregating features from both networks. This work also utilize the "relative difficulty" that is proposed in (Lee et al., 2021). We reproduced the results utilizing the official code offered by the authors. The loss function proposed in this work is as follows:

$$\mathcal{L}_{\text{total}} = \mathcal{L}_{\text{dis}} + \lambda_{\text{swap}}\mathcal{L}_{\text{swap}},$$

where

$$\mathcal{L}_{\text{swap}} = W(z)\mathcal{L}_{\text{CE}}(C_d(z_{\text{swap}}, y) + \lambda_{\text{swap}_b}\mathcal{L}_{\text{GCE}}(C_b(z_{\text{swap}}, \tilde{y}))$$
$$\mathcal{L}_{\text{dis}} = W(z)\mathcal{L}_{\text{CE}}(C_d(z, y) + \lambda_{\text{dis}}\mathcal{L}_{\text{GCE}}(C_b(z, y)),$$
$$W(z) = \frac{\mathcal{L}_{\text{CE}}(C_b(z), y)}{\mathcal{L}_{\text{CE}}(C_b(z), y) + \mathcal{L}_{\text{CE}}(C_d(z), y)}.$$

Except for the swapped feature, $z_{\text{swap}}$ all terms are identical to those in LfF explanation.

(4) GEORGE (Sohoni et al., 2020) aims to debias by measuring and mitigating hidden stratification without requiring access to subclass labels. Assume there are $n$ data points $x_1, ..., x_n \in \chi$ and associated superclass (target) labels $y_1, ..., y_n \in \{1, \cdots, C\}$. Furthermore, each datapoint $x_i$ is associated with a latent (unobserved) subclass label $z_i$. George consists of three steps. The author trains the biased model using ERM. Next, to estimate an approximate subclass (latent) label, apply UMAP dimensionality reduction (McInnes et al., 2018) to the features of a given training dataset at the ERM model. Here, GEORGE cluster the output of the reduced dimension for the data of each superclass into $K$ clusters, where $K$ is chosen automatically. The original paper contains a detailed

description of the clustering process. Lastly, to improve performance on these estimated subclasses, GEORGE minimizes the maximum per-cluster average loss (*i.e.*, $(x, y) \sim \hat{P}_{\tilde{z}}$), by using the cluster as groups in the G-DRO objective (Sagawa et al., 2019). The loss function proposed in this work is as follows:

$$\underset{L, f_\theta}{\text{minimize}} \ \underset{1 \leq \tilde{z} \leq K}{\max} \ \mathbb{E}_{(x,y) \sim \hat{P}_{\tilde{z}}}[l(L \circ f_\theta(x), y)]$$

where $f_\theta$ and $L$ are parameterized feature extractor and classifier, respectively.

(5) BPA (Seo et al., 2022) aims to debias by using the technique of feature clustering and cluster reweighting. It consists of three steps. First, the author trains the biased model based on ERM. Next, at the biased model, cluster all training samples into $K$ clusters based on the feature, where $K$ is the hyperparameter. Here, $h(x, y; \tilde{\theta}) \in \mathcal{K} = \{1, \cdots, K\}$ denote the cluster mapping function of data $(x, y)$ derived by the biased model with parameter $\tilde{\theta}$. At the last step, BPA computes the proper importance weight, $w_k$ for the $k$-th cluster, where $k \in \mathcal{K}$ and the final objective of debiasing the framework is given by minimizing the weighted empirical risk as follows:

$$\underset{\theta}{\text{minimize}} \left\{ \mathbb{E}_{(x,y) \sim P} \left[ w_{h(x,y;\tilde{\theta})}(\theta) l(x, y; \theta) \right] \right\},$$

Concretely, for any iteration number $T$, the momentum method based on the history set $\mathcal{H}_T$, which is defined as:

$$\mathcal{H}_T = \left\{ 1 \leq t \leq T \mid \frac{\mathbb{E}_{(x,y) \sim P_k} [l((x, y); \theta_t)]}{N_k} \right\},$$

where $N_k$ is the number of the data belonging to $k$-th cluster.

(6) CNC (Zhang et al., 2022b) aims to debias by learning representation such that samples in the same class are close but different groups are far. CNC is composed of two steps: (1) inferring pseudo group label, (2) supervised contrastive learning. Get the ERM-based model $f$ and the pseudo prediction $\hat{y}$ first, then standard argmax over the final layer outputs of the model $f$. Next, CNC trains the debiased model based on supervised contrastive learning using pseudo prediction $\hat{y}$. The detailed process of contrastive learning for each iteration is as follows:

- From the selected batch, sample the one anchor data (x,y).
- Construct the set of positives samples $\{(x_m^+, y_m^+)\}$ which is belong to the batch, satisfying $y_m^+ = y$ and $\hat{y}_m^+ \neq \hat{y}$.
- Similarly, construct the set of negative samples $\{(x_n^-, y_n^-)\}$ which is belong to the batch, satisfying $y_n^- \neq y$ and $\hat{y}_n^- = \hat{y}$.
- With the loss of generality, assume the cardinality of the positive and negative sets are $M$ and $N$, respectively.
- Weight update based on the gradient of the loss function $\hat{L}(f_\theta; x, y)$, the detail is like below:

$$\hat{L}(f_\theta; x, y) = \lambda \hat{L}_{\text{con}}^{\sup}(x, \{x_m^+\}_{m=1}^M, \{x_n^-\}_{n=1}^N; f_{\text{enc}}) + (1 - \lambda) \hat{L}_{\text{cross}}(f_\theta; x, y).$$

  Here, $\lambda \in [0, 1]$ is a hyperparameter and $\hat{L}_{\text{cross}}(f_\theta; x, y)$ is an average cross-entropy loss over $x$, the $M$ positives, and $N$ negatives. Moreover, $f_{\text{enc}}$ is the feature extractor part of $f_\theta$ and the detail formulation of $\hat{L}_{\text{con}}^{\sup}(x, \{x_m^+\}_{m=1}^M, \{x_n^-\}_{n=1}^N; f_{\text{enc}})$ is like below:

$$-\frac{1}{M} \sum_{r=1}^M \log \frac{\exp(f_{\text{enc}}(x)^T f_{\text{enc}}(x_r^+)/\tau)}{\sum_{m=1}^M \exp(f_{\text{enc}}(x)^T f_{\text{enc}}(x_m^+)/\tau) + \sum_{n=1}^N \exp(f_{\text{enc}}(x)^T f_{\text{enc}}(x_n^-)/\tau)}.$$

(7) EIIL (Creager et al., 2021) proposes a novel invariant learning framework that does not require prior knowledge of the environment. EIIL is composed of three steps: (i) training based on ERM, (ii) environment inference (EI), and (iii) invariant learning (IL). In the first step, EIIL gets a biased model by minimizing ERM. Next, based on the feature extractor trained in (i), optimize the EI objective to infer environments from each training dataset. The object of EI is to sort training examples that maximally separate the spurious features so that they facilitate effective invariant learning. Lastly, to get a debiased model, optimize the classifier and feature extractor by minimizing the invariant learning objective.

## B EXPERIMENT DETAILS AND ADDITIONAL ANALYSIS

### B.1 SETTINGS

This section discusses how our experiment was set up, including architecture, image processing, and implementation details.

**Architecture.** For the colored MNIST, we use simple convolutional networks consisting of three CNN layers with kernel size 4 and channel sizes {8, 32, 64} for each layer. Also, we utilize average pooling at the end of each layer. Batch normalization and dropout techniques are used for regularization. Detailed network configurations are below. Similarly, for the multi-bias MNIST, we use four CNN layers with kernel size {7, 7, 5, 3} and channel size {8, 32, 64, 128}, respectively. For corrupted CIFAR, BAR, and BFFHQ, we utilize ResNet-18, which is provided by the open-source library torchvision. For CelebA, we follow the experimental setting of CNC (Zhang et al., 2022b), which uses ResNet-50 as a backbone network. For CivilCOmments-WILDS, we use pretrained-BRET for the backbone network and exactly the same hyperparameters as in (Zhang et al., 2022b).

**SimConv-1.**

```
(conv1): Conv2d(3, 8, kernel_size=(4, 4), stride=(1, 1))
(bn1): BatchNorm2d(8, eps=1e-05, momentum=0.1, affine=True,
track_running_stats=True)
(relu1): ReLU()
(dropout1): Dropout(p=0.5, inplace=False)
(avgpool1): AvgPool2d(kernel_size=2, stride=2, padding=0)
(conv2): Conv2d(8, 32, kernel_size=(4, 4), stride=(1, 1))
(bn2): BatchNorm2d(32, eps=1e-05, momentum=0.1, affine=True,
track_running_stats=True)
(relu2): ReLU()
(dropout2): Dropout(p=0.5, inplace=False)
(avgpool2): AvgPool2d(kernel_size=2, stride=2, padding=0)
(conv3): Conv2d(32, 64, kernel_size=(4, 4), stride=(1, 1))
(relu3): ReLU()
(bn3): BatchNorm2d(64, eps=1e-05, momentum=0.1, affine=True,
track_running_stats=True)
(dropout3): Dropout(p=0.5, inplace=False)
(avgpool3): AdaptiveAvgPool2d(output_size=(1, 1))
(fc): Linear(in_features=64, out_features=$num_class, bias=True)
```

**SimConv-2.**

```
(conv1): Conv2d(3, 8, kernel_size=(7, 7), stride=(1, 1))
(bn1): BatchNorm2d(8, eps=1e-05, momentum=0.1, affine=True,
track_running_stats=True)
(relu1): ReLU()
(dropout1): Dropout(p=0.5, inplace=False)
(avgpool1): AvgPool2d(kernel_size=3, stride=3, padding=0)
(conv2): Conv2d(8, 32, kernel_size=(7, 7), stride=(1, 1))
(bn2): BatchNorm2d(32, eps=1e-05, momentum=0.1, affine=True,
track_running_stats=True)
(relu2): ReLU()
(dropout2): Dropout(p=0.5, inplace=False)
(avgpool2): AvgPool2d(kernel_size=3, stride=3, padding=0)
(conv3): Conv2d(32, 64, kernel_size=(5, 5), stride=(1, 1))
(relu3): ReLU()
(bn3): BatchNorm2d(64, eps=1e-05, momentum=0.1, affine=True,
track_running_stats=True)
(dropout3): Dropout(p=0.5, inplace=False)
(conv4): Conv2d(64, 128, kernel_size=(3, 3), stride=(1, 1))
(relu4): ReLU()
(bn4): BatchNorm2d(128, eps=1e-05, momentum=0.1, affine=True,
```

```
track_running_stats=True)
(dropout4): Dropout(p=0.5, inplace=False)
(avgpool): AdaptiveAvgPool2d(output_size=(1, 1))
(fc): Linear(in_features=128, out_features=$num_class, bias=True)
```

## B.2 PGD IMPLEMENTATION DETAILS

**Image processing** We train and evaluate with a fixed image size. For colored MNIST case ($28 \times 28$), multi-bias MNIST ($56 \times 56$), corrupted CIFAR ($32 \times 32$), and the remains ($224 \times 224$). For the CMNIST, MBMNIST, CCIFAR, BAR, and BFFHQ, we utilize random resize crop, random rotation, and color jitter to avoid overfitting. We use normalizing with a mean of $(0.4914, 0.4822, 0.4465)$, and standard deviation of $(0.2023, 0.1994, 0.2010)$ for CCIFAR, BAR, and BFFHQ cases.

**Implementation** For table 1 and Table 2 reported in Setion 4, we reproduce all experimental results referring to other official repositories: [7] [8] [9] [10]. The differences compared to the baseline codes are network architecture for CMNIST and usage of data augmentation. Here, we use the same architecture for CMNIST and data augmentation for all algorithms for a fair comparison. Except for JTT, all hyperparameters for CCIFAR and BFFHQ follow previously reported parameters in repositories. We grid-search for other cases, MNIST variants, and BAR. We set the only hyperparameter of PGD, $\alpha = 0.7$, as proposed by the original paper (Zhang & Sabuncu, 2018). A summary of the hyperparameters that we used is reported in Table 6.

| | Colored MNIST | Multi-bias MNIST | Corrupted CIFAR | BAR | Biased FFHQ | CelebA | CivilComments-WILDS |
|---|---|---|---|---|---|---|---|
| Optimizer | SGD | SGD | Adam | SGD | Adam | Adam | SGD |
| Batch size | 128 | 32 | 256 | 16 | 64 | 256 | 16 |
| Learning rate | 0.02 | 0.01 | 0.001 | 0.0005 | 0.0001 | 0.0001 | 0.00001 |
| Weight decay | 0.001 | 0.0001 | 0.001 | 1e-5 | 0.0 | 0.01 | 0.01 |
| Momentum | 0.9 | 0.9 | - | 0.9 | - | - | 0.9 |
| Lr decay | 0.1 | 0.1 | 0.5 | 0.1 | 0.1 | Cosine annealing | 0.1 |
| Decay step | 40 | - | 40 | 20 | 32 | - | - |
| Epoch | 100 | 100 | 200 | 100 | 160 | 100 | 5 |
| GCE $\alpha$ | 0.7 | 0.7 | 0.7 | 0.7 | 0.7 | 0.7 | 0.7 |

Table 6: Hyperparameter details

For Table 3 reported in Section 4, we follow the implementation settings of CelebA and CivilComments-WILDS, suggested by Seo et al. (2022) and Liu et al. (2021), respectively. A summary of the hyperparameters that we used is reported in Table 6. We conduct our experiments mainly using a single Titan XP GPU for all cases.

---

[7] https://github.com/alinlab/LfF

[8] https://github.com/clovaai/rebias

[9] https://github.com/kakaoenterprise/Learning-Debiased-Disentangled

[10] https://github.com/anniesch/jtt

## C    CASE STUDIES ON PGD

In this section, we analyze PGD in many ways. Most analyses are based on the CMNIST dataset, and the experimental setting is the same as the existing setting unless otherwise noted. For example, all experiments used the same data augmentation, color jitter, resize crop, and random rotation.

### C.1    STUDY 1: ABLATION STUDY ON GCE PARAMETER $\alpha$

The only hyper-parameter used in PGD is the GCE parameter $\alpha$. We experimented with this value at $0.7$ according to the protocol of LfF (Nam et al., 2020). However, we need to compare the various cases of $\alpha$. To analyze this, we run PGD with various $\alpha$ and report the performance in Table 7. As in Table 7, the debiased model performs best when the GCE parameter is 0.9. This is because the biased model is fully focused on the bias feature, rather than the target feature, which can be seen from the unbiased test accuracy of the biased model, as in the bottom of Table 7.

| Colored MNIST | $\alpha = 0.3$ | $\alpha = 0.5$ | $\alpha = 0.7$ | $\alpha = 0.9$ |
|---|---|---|---|---|
| | Debiased model | | | |
| $\rho = 0.5\%$ | $89.93_{\pm 0.19}$ | $94.70_{\pm 0.23}$ | $96.88_{\pm 0.28}$ | $96.79_{\pm 0.04}$ |
| $\rho = 1\%$ | $96.32_{\pm 0.12}$ | $97.27_{\pm 0.16}$ | $97.35_{\pm 0.12}$ | $97.59_{\pm 0.04}$ |
| $\rho = 5\%$ | $98.80_{\pm 0.05}$ | $98.82_{\pm 0.02}$ | $98.62_{\pm 0.14}$ | $98.78_{\pm 0.02}$ |
| | Biased model | | | |
| $\rho = 0.5\%$ | $19.86_{\pm 0.93}$ | $18.70_{\pm 0.89}$ | $18.12_{\pm 0.09}$ | $17.39_{\pm 0.86}$ |
| $\rho = 1\%$ | $22.40_{\pm 2.12}$ | $21.04_{\pm 1.54}$ | $19.71_{\pm 0.24}$ | $19.12_{\pm 1.12}$ |
| $\rho = 5\%$ | $53.24_{\pm 1.18}$ | $49.46_{\pm 1.13}$ | $43.97_{\pm 1.75}$ | $39.40_{\pm 2.65}$ |

Table 7: Ablation study on GCE parameter $\alpha$.

### C.2    STUDY 2: MULTI-STAGE VS SINGLE-STAGE

PGD computes the per-sample gradient norm only once, between training the biased model and the debiased model. However, an update of the per-sample gradient can be performed repeatedly at each epoch (i.e., single-stage). In other words, PGD can be modeled to run the

|  | Training time | | Test Acc. | |
|---|---|---|---|---|
| Colored MNIST | Single-stage | Multi-stage | Single-stage | Multi-stage |
| $\rho = 0.5\%$ | 2h 53m 40s | 33m 39s | $92.19_{\pm 0.12}$ | $96.88_{\pm 0.28}$ |
| $\rho = 1\%$ | 2h 54m 45s | 32m 28s | $97.23_{\pm 0.34}$ | $97.35_{\pm 0.12}$ |
| $\rho = 5\%$ | 2h 49m 31s | 34m 13s | $98.44_{\pm 0.17}$ | $98.62_{\pm 0.14}$ |

Table 8: Multi-stage vs Single-stage

following loop: updating the biased model, updating the sampling probability, and updating the debiased model. In this section, we justify why we use the multi-stage approach. We report the performance of multi-stage and single-stage PGD on the colored MNIST dataset. As in Table 8, the single-stage method has two characteristics: (1) it requires more training time than the multi-stage method. (2) It has lower unbiased accuracy compared to the multi-stage method. The longer training time is due to the high computational resources required to compute the per-sample gradient norm. Moreover, because the single-stage method's sampling probability changes the training distribution over epochs, the debiased model suffers from unstable training and loses debiasing performance.

### C.3    STUDY 3: RESAMPLING VS REWEIGHTING

To support our algorithm design, we provide further experimental analysis, i.e., resampling versus reweighting. Reweighting (Nam et al., 2020; Lee et al., 2021) and resampling (Liu et al., 2021) are the two main techniques to debias by up-weighting bias-conflicting samples. PGD is an algorithm that modifies the sampling probability by

| Colored MNIST | Resampling | Reweighting |
|---|---|---|
| $\rho = 0.5\%$ | $96.88_{\pm 0.28}$ | $94.70_{\pm 0.4}$ |
| $\rho = 1\%$ | $97.35_{\pm 0.12}$ | $97.20_{\pm 0.1}$ |
| $\rho = 5\%$ | $98.62_{\pm 0.14}$ | $98.51_{\pm 0.03}$ |

Table 9: Reweighting vs resampling

using the per-sample gradient norm. To check whether PGD works with reweighting, we examine the results of PGD with reweighting on colored MNIST dataset and report them in Table 9. We compute the weight for each sample as follows: $w(x_i, y_i) = |\mathcal{D}_n| \times \frac{\|\nabla_\theta \mathcal{L}_{\text{CE}}(x_i, y_i; \theta_b)\|_2}{\sum_{(x_i, y_i) \in \mathcal{D}_n} \|\nabla_\theta \mathcal{L}_{\text{CE}}(x_i, y_i; \theta_b)\|_2}$.

As in Table 9, PGD with resampling slightly outperforms PGD with reweighting. As argued in (An et al., 2020), this gain from resampling can be explained by the argument that resampling is more stable and better than reweighting.

## C.4 STUDY 4: ANALYSIS OF THE PURE EFFECT OF GRADIENT-NORM-BASED SCORE

PGD performance improvement comes not only from the two-stage and resampling modules that we wrote about in Appendix C.2 and Appendix C.3, but also from the gradient score. To verify this, we report the following two results: (1) resample based on per-sample loss rather than per-sample gradient norm in PGD, and (2) change the relative difficulty score of LfF to gradient norm. As shown in Table 10, we can conclude the following two results: (1) loss of the last epoch of the first stage in a two-stage approach is not suitable for resampling, and (2) the results of replacing the relative difficulty metric of LfF with a gradient norm show that the gradient norm has better discriminative properties for bias-conflicting samples than loss.

| Method | $\rho = 0.5\%$ | $\rho = 1\%$ | $\rho = 5\%$ |
|---|---|---|---|
| LfF | $91.35_{\pm 1.35}$ | $96.88_{\pm 0.2}$ | $98.18_{\pm 0.05}$ |
| PGD + Loss | $30.63_{\pm 2.23}$ | $34.04_{\pm 3.00}$ | $78.48_{\pm 1.41}$ |
| LfF + Gradient | $93.29_{\pm 0.39}$ | $97.55_{\pm 0.24}$ | $98.37_{\pm 0.20}$ |
| PGD | $96.88_{\pm 0.28}$ | $98.35_{\pm 0.12}$ | $98.62_{\pm 0.14}$ |

Table 10: loss score vs gradient norm score

## C.5 STUDY 5: COMPUTATION COST

Debiasing algorithms require an additional computational cost. To evaluate the computational cost of PGD, we report the training time in Table 11. We conduct this experiment by using the colored MNIST with $\rho = 0.5\%$. As in the top of Table 11, we report the training time of four methods: vanilla, LfF, Disen, and PGD. Here, PGD spends a longer amount of training time. This is because there is no module for computing per-sample gradient in a batch manner. At the bottom of Table 11, we report part-by-part costs to see which parts consume the most time. Note that Steps 1, 2, and 3 represent training the biased model, computing the per-sample gradient-norm, and training the debiased model, respectively. We can conclude with the following two facts. (1) Step 2 (computing the per-sample gradient norm and sampling probability) spends $4.3\%$ of training time. (2) Resampling based on the modified sampling probability $h(x)$ requires an additional cost of 1m 27s by seeing the difference between the computing times of Step 3 and Step 1.

| | Vanilla | LfF | Disen | ours |
|---|---|---|---|---|
| Computation time | 14m 59s | 21m 35s | 23m 18s | 33m 31s |
| | Step 1 | Step 2 | Step 3 | |
| Computation time | 15m 19s | 1m 26s | 16m 46s | |

Table 11: Computation cost

## C.6 STUDY 6: WHAT IF PGD RUNS ON UNBIASED DATASETS

We examine whether PGD fails when an unbiased dataset is given. To verify this, we report two types of additional results: (1) unbiased CMNIST (*i.e.,* $\rho = 90\%$) and (2) conventional public dataset (*i.e.,* CIFAR10). We follow the experimental setting of CMNIST for the unbiased CMNIST case. On the other hand, we train ResNet18 (He et al., 2016) for CIFAR10 with the SGD optimizer, 0.9 momentum, $5e - 4$ weight decay, 0.1 learning rate, and Cosine Annealing LR decay scheduler. As shown in Table 12, PGD does not suffer significant performance degradation in unbiased CMNIST. Furthermore, it performs better than the vanilla model on the CIFAR10 dataset. This means that the training distribution that PGD changes do not cause significant performance degradation. In other words, PGD works well, regardless of whether the training dataset is balanced or unbiased.

| Vanilla | LfF | Disen | Ours |
|---|---|---|---|
| $\rho = 90\%$ Colored MNIST | | | |
| $99.04_{\pm 0.05}$ | $98.75_{\pm 0.07}$ | $99.31_{\pm 0.1}$ | $98.43_{\pm 0.11}$ |
| Natrual CIFAR10 | | | |
| $94.24_{\pm 0.01}$ | - | - | $94.79_{\pm 0.02}$ |

Table 12: Results on unbiased CMNIST and natural CIFAR10 cases.

## D    EMPIRICAL EVIDENCE OF PGD

As same with the setting of Appendix C, in this section, we also use the existing CMNIST setting, such as data augmentation, hyperparameters.

### D.1    CORRELATION BETWEEN GRADIENT NORM AND BIAS-ALIGNMENT OF THE CMNIST

To check if the per-sample gradient norm efficiently separates the bias-conflicting samples from the bias-aligned samples, we plot the gradient norm distributions of the colored MNIST (CMNIST). For comparison, we normalized the per-sample gradient norm as follows: $\frac{\|\nabla_\theta \mathcal{L}_{\text{CE}}(x_i,y_i;\theta_b)\|}{\max_{(x_i,y_i)\in\mathcal{D}_n}\|\nabla_\theta \mathcal{L}_{\text{CE}}(x_i,y_i;\theta_b)\|}$. As in Figure 9, the bias-aligned sample has a lower gradient norm (blue bars) than the bias-conflicting samples (red bars).

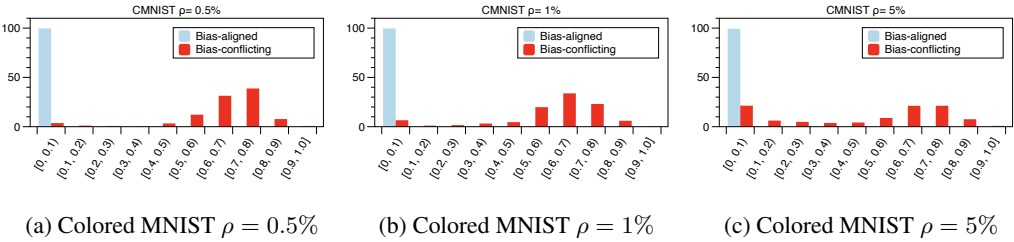

(a) Colored MNIST $\rho = 0.5\%$       (b) Colored MNIST $\rho = 1\%$       (c) Colored MNIST $\rho = 5\%$

Figure 9: Histogram of per-sample gradient norm.

### D.2    PGD DOES NOT LEARN ONLY THE SECOND-EASIEST FEATURE

We provide the results of the following experimental settings: The target feature is color, and the bias feature is digit shape, i.e., the task is to classify the color, not the digit shape. Let us give an example of this task. When one of the target classes is red, this class is aligned with one of the digits (e.g., "0"). In other words, the bias-aligned samples in this class are (Red, "0"), and the bias-conflicting samples are (e.g., (Red, "1"), (Red, "2"),.., (Red, "9")). Note that, as shown in LfF (Nam et al., 2020), color is empirically known to be easier to learn than digit shape; we think that the above scenario reflects the concern: whether PGD is only targeting the second-easiest feature (digit shape). Therefore, if the concern is correct, PGD may fail in this color target MNIST scenario since the model will learn digit shape. However, as shown in the table below, vanilla, PGD, and LfF perform well in that case.

|  | Vanilla (Digit) | **Vanilla (Color)** | LfF (Digit) | **LfF (Color)** | PGD (Digit) | **PGD (Color)** |
|---|---|---|---|---|---|---|
| $\rho = 0.5\%$ | 60.94 | **90.33** | 91.35 | **91.16** | 96.88 | **98.92** |
| $\rho = 1\%$ | 79.13 | **92.53** | 96.88 | **96.12** | 98.35 | **99.58** |
| $\rho = 5\%$ | 95.12 | **96.96** | 98.18 | **99.11** | 98.62 | **99.7** |

Table 13: Digit target MNIST vs Color target MNIST

We can also support this result by seeing the distribution of the normalized gradient norms, $\|\nabla_\theta\mathcal{L}_{\text{CE}}(x_i,y_i;\theta_b)\|_2/\max_{(x_i,y_i)\in\mathcal{D}_n}\|\nabla_\theta\mathcal{L}_{\text{CE}}(x_i,y_i;\theta_b)\|_2 \in [0,1]$, extracted from the biased model $\theta_b$ (trained in Step 1 of Algorithm 1 of Section 3).

|  | [0.0,0.1) | [0.1,0.2) | [0.2,0.3) | [0.3,0.4) | [0.4,0.5) | [0.5,0.6) | [0.6,0.7) | [0.7,0.8) | [0.8,0.9) | [0.9,1.0] |
|---|---|---|---|---|---|---|---|---|---|---|
| Bias-aligned | 53504 | 88 | 40 | 21 | 13 | **18** | **16** | **17** | **8** | **4** |
| Bias-conflicting | 212 | 18 | 7 | 10 | 8 | **6** | **5** | **4** | **0** | **1** |

Table 14: Number of samples at each bin: Color target MNIST ($\rho = 0.5\%$)

The numbers filled in Table 14 are the number of data items belonging to each bin category. We can check that *there are no bias-conflicting samples whose gradient norm is significantly larger than the bias-aligned samples.* In other words, PGD *does not force the debiased model to learn the digit shape (i.e., the second-easiest feature) in this scenario.* This scenario brings similar performance to Vanilla.

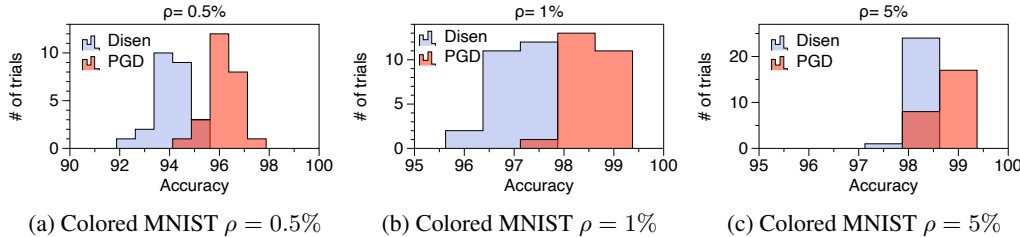

(a) Colored MNIST $\rho = 0.5\%$     (b) Colored MNIST $\rho = 1\%$     (c) Colored MNIST $\rho = 5\%$

Figure 10: Histogram of unbiased test accuracy among 25 trials for each.

### D.3 HISTOGRAM OF THE RESULTS OF 25 TRIALS IN CMNIST

For in-depth analysis, we provide the results of 25 tests on CMNIST in Figure 10. We compare with Disen, which shows the best performance except for PGD. However, very few cases overlap, as shown in Figure 10. We conduct a t-test for a more in-depth analysis of this. When the alternative hypothesis is established that PGD is superior to Disen, the p-value for it has values of 0.01572, 0.01239, and 0.29370, respectively. In other words, it can be said that as the bias becomes worse, the superiority of PGD stands out.

## E ADDITIONAL RESULTS ON OTHER DATASETS

### E.1 CELEBA WITH RESNET18 SETTING.

|  |  | Vanilla | LfF | GEORGE | BPA | Ours |
|---|---|---|---|---|---|---|
| CelebA[†] | Avg. | 80.52 | 84.89 | 83.13 | 90.18 | 89.27 |
|  | Worst | 41.02 | 57.96 | 65.45 | 82.54 | **82.73** |

Table 15: Average and worst test accuracy with CelebA setting of (Seo et al., 2022). The results of comparison algorithms for CelebA [†] are the results reported in (Seo et al., 2022). The best worst accuracy is indicated in **bold**.

We reported the results of CelebA in Table 3 of section 4, following the settings of (Zhang et al., 2022b). For comparison with more diverse algorithms, we further report the CelebA results according to the settings of (Seo et al., 2022). Note that the (Zhang et al., 2022b) and (Seo et al., 2022) used a different model, each using ResNet50 and ResNet18, respectively. As in Table 15, PGD shows competitive performance among the other baselines.

# F  BACKGROUNDS FOR THEORETICAL ANALYSIS

## F.1  NOTATIONS SUMMARY.

For convenience, we describe notations used in Section 5, Appendix F, G, and H.

Table 16: Notation Table

| Notation | Description | Remark |
|---|---|---|
| | Variables | |
| $(x, y)$ | (image, label) | $x \in \mathbb{R}^d, y \in C = \{1, ..., c\}$ |
| $y_{true}(x)$ | the true label of image $x$ | labeled by the oracle model $f(y|x, \theta^\star)$ |
| $\theta$ | model parameter | - |
| $\theta^\star$ | oracle model parameter | satisfying $f(y_{true}(x)|x, \theta^\star) = 1$ for any $x$ |
| $\mathcal{D}_n$ | training dataset | composed of $\{(x_i, y_i)\}_{i=1}^n \sim p(x, y|\theta^\star)$ |
| $h(x)$ | sampling probability of each sample in $\mathcal{D}_n$ | satisfying $h(x) \in \mathcal{H}$ |
| | Distributions | |
| $\mathcal{P}(x)$ | general distribution of input image $x$ | - |
| $p(x)$ | distribution of training image | - |
| $q(x)$ | distribution of test image | - |
| $f(y|x, \theta)$ | conditional distribution with model parameter $\theta$ | - |
| $\mathcal{P}(x, y|\theta)$ | general joint distribution with model parameter $\theta$ | $\mathcal{P}(x)f(y|x, \theta)$ |
| $p(x, y|\theta^\star)$ | joint distribution of the training dataset | $p(x)f(y|x, \theta^\star)$ |
| $q(x, y|\theta^\star)$ | joint distribution of the test dataset | $q(x)f(y|x, \theta^\star)$ |
| | Estimators | |
| $\hat{\theta}_{h(x), \mathcal{D}_n}$ | MLE solution on the $\mathcal{D}_n$ with $h$ | $\triangleq \arg\max_\theta \quad \sum_{i=1}^n h(x_i) \log f(y_i|x_i, \theta)$ |
| $\hat{\theta}_{U(x), \mathcal{D}_n}$ | MLE solution on the $\mathcal{D}_n$ with uniform distribution $U$ | solution of ERM |
| | Fisher Information | |
| $I_{\mathcal{P}(x)}(\theta)$ | Fisher Information | $\mathbb{E}_{(x,y) \sim \mathcal{P}(x)f(y|x,\theta)}[\nabla_\theta \log f(y|x, \theta) \nabla_\theta^\top \log f(y|x, \theta)]$ |
| $\hat{I}_{h(x)}(\theta)$ | Empirical Fisher Information | $\sum_{i=1}^n h(x_i) \nabla_\theta \log f(y_i|x_i, \theta) \nabla_\theta^\top \log f(y_i|x_i, \theta)$ |
| | Set | |
| $\mathcal{H}$ | set of all possible $h(x)$ on $\mathcal{D}_n$ | $\{h(x)| \sum_{(x_i,y_i) \in \mathcal{D}_n} h(x_i) = 1 \text{ and } h(x_i) \geq 0 \quad \forall (x_i, y_i) \in \mathcal{D}_n\}$ |
| $\mathcal{M}$ | set of all possible marginal $\mathcal{P}(x)$ on input space $\mathcal{X}$ | $\{\mathcal{P}(x)| \int_{x \in \mathcal{X}} \mathcal{P}(x)\, dx = 1\}$ |
| $\mathcal{W}$ | set of all possible $(x, y_{true}(x))$ | - |
| $supp(\mathcal{P}(x, y|\theta))$ | Support set of $\mathcal{P}(x, y|\theta)$ | $\{(x, y) \in X \times \{1, \cdots, c\} \mid \mathcal{P}(x, y|\theta) \neq 0\}, \ \forall \ \mathcal{P}(x, y|\theta)$ |
| | Order notations in probability | |
| $O_p$ | Big $O$, stochastic boundedness | - |
| $o_p$ | Small $o$, convergence in probability | - |
| | Toy example (Appendix H) | |
| $M$ | set of majority (*i.e.,* bias-aligned) samples | - |
| $m$ | set of minority (*i.e.,* bias-conflicting) samples | - |
| $g_M(\theta)$ | gradient of samples in $M$ at $\theta$ | - |
| $g_m(\theta)$ | gradient of samples in $m$ at $\theta$ | - |
| $h_M^\star(x)$ | Optimal sampling probability of samples in $M$ | - |
| $h_m^\star(x)$ | Optimal sampling probability of samples in $m$ | - |

## F.2  MAIN ASSUMPTION

Here, we organize the assumptions that are used in the proof of Theorems. These are basically used when analyzing models through Fisher information. The assumptions are motivated by (Sourati et al., 2016).

**Assumption 1.**

(A0). The general joint distribution $\mathcal{P}(x, y|\theta)$ is factorized into the conditional distribution $f(y|x, \theta)$ and the marginal distribution $\mathcal{P}(x)$, not depend on model parameter $\theta$, that is:

$$\mathcal{P}(x, y|\theta) = \mathcal{P}(x)f(y|x, \theta). \tag{2}$$

Thus, the joint distribution is derived from model parameter $\theta$ and the marginal distribution $\mathcal{P}(x)$, which is determined from the task that we want to solve. Without loss of generality, we match the joint distribution's name with the marginal distribution.

(A1). *(Identifiability):* The CDF $\mathcal{P}_\theta$ (whose density is given by $\mathcal{P}(x, y|\theta)$) is identifiable for different parameters. Meaning that for every distinct parameter vectors $\theta_1$ and $\theta_2$ in $\Omega$, $\mathcal{P}_{\theta_1}$ and $\mathcal{P}_{\theta_2}$ are also distinct. That is,

$$\forall \theta_1 \neq \theta_2 \in \Omega, \quad \exists A \subseteq X \times \{1, \cdots, c\} \quad \text{s.t.} \quad \mathcal{P}_{\theta_1}(A) \neq \mathcal{P}_{\theta_2}(A),$$

where $X$, $\{1, \cdots, c\}$ and $\Omega$ are input, label, and model parameter space, respectively.

(A2). The joint distribution $\mathcal{P}_\theta$ has common support for all $\theta \in \Omega$.

(A3). *(Model Faithfulness)*: For any $x \in X$, we assume an oracle model parameter $\theta^\star$ that generates $y_{\text{true}}(x)$, a true label of input $x$ with the conditional distribution $f(y_{\text{true}}(x)|x, \theta^\star) = 1$.

(A4). *(Training joint):* Let $p(x)$ denote the training marginal with no dependence on the parameter. Then, the set of observations in $\mathcal{D}_n \triangleq \{(x_1, y_1). \cdots (x_n, y_n)\}$ are drawn independently from the training/proposal joint distribution of the form $p(x, y|\theta^\star) \triangleq p(x)f(y|x, \theta^\star)$, because we do not think the existence of mismatched label data situation in the training data.

(A5). *(Test joint):* Let $q(x)$ denote the test marginal without dependence on the parameter. The unseen test pairs are distributed according to the test/true joint distribution of the form $q(x, y|\theta^\star) \triangleq q(x)f(y|x, \theta^\star)$, because we do not think the existence of mismatched label data situation in the test task.

(A6). *(Differentiability):* The log-conditional $\log f(y|x, \theta)$ is of class $\mathcal{C}^3(\Omega)$ for all $(x, y) \in X \times \{1, 2, \cdots, c\}$, when being viewed as a function of the parameter.[11]

(A7). The parameter space $\Omega$ is compact, and there exists an open ball around the true parameter of the model $\theta^\star \in \Omega$.

(A8). *(Invertibility):* The arbitrary Fisher information matrix $I_{\mathcal{P}(x)}(\theta)$ is positive definite and therefore invertible for all $\theta \in \Omega$.

(A9). $\{(x, y) \in supp(q(x, y|\theta^\star)) \mid \nabla^2_\theta \log q(x, y|\theta^\star) \text{ is singular}\}$ is a measure zero set.

In contrast to (Sourati et al., 2016), we modify (A3) so that the oracle model always outputs a hard label, *i.e.,* $f(y_{true}(x)|x, \theta^\star) = 1$ and add (A9) which is not numbered but noted in the statement of Theorem 3 and Theorem 11 in (Sourati et al., 2016).

## F.3 PRELIMINARIES

We organize the two types of background knowledge, maximum likelihood estimator (MLE) and Fisher information (FI), needed for future analysis.

### F.3.1 MAXIMUM LIKELIHOOD ESTIMATOR (MLE)

In this section, we derive the maximum likelihood estimator in a classification problem with sampling probability $h(x)$. Unless otherwise specified, training set $\mathcal{D}_n = \{(x_i, y_i)\}_{i=1}^n$ is sampled from $p(x, y|\theta^\star)$. For given probability mass function (PMF) $h(x)$ on $\mathcal{D}_n$, we define MLE $\hat{\theta}_{h(x), \mathcal{D}_n}$ as follows:

$$
\begin{aligned}
\hat{\theta}_{h(x), \mathcal{D}_n} &\triangleq \arg\max_\theta \quad \log \mathbb{P}(\mathcal{D}_n|\theta; h(x)) \\
&= \arg\min_\theta \quad -\sum_{i=1}^n h(x_i) \log p(x_i, y_i|\theta) \quad (3) \\
&= \arg\min_\theta \quad -\sum_{i=1}^n h(x_i) \log f(y_i|x_i, \theta) \quad (4) \\
&= \arg\min_\theta \quad \sum_{i=1}^n h(x_i)\, \mathcal{L}_{\text{CE}}(x_i, y_i; \theta). \quad (5)
\end{aligned}
$$

In (3) and (4), (A0) and (A4) of Assumption 1 in Appendix F were used, respectively. It is worth noting that MLE $\hat{\theta}_{h(x), \mathcal{D}_n}$ is a variable that is influenced by two factors: (1) a change in the training

---

[11]We say that a function $f : X \to Y$ is of $\mathcal{C}^p(X)$, for an integer $p > 0$, if its derivatives up to $p$-th order exist and are continuous at all points of $X$.

dataset $\mathcal{D}_n$ and (2) the adjustment of the sampling probability $h(x)$. If $h(x)$ is a uniform distribution $U(x)$, then the result of empirical risk minimization (ERM) is $\hat{\theta}_{U(x),\mathcal{D}_n}$.

### F.3.2 FISHER INFORMATION (FI)

**General definition of FI.** Fisher information (FI), denoted by $I_{\mathcal{P}(x)}(\theta)$, is a measure of sample information from a given distribution $\mathcal{P}(x,y|\theta) \triangleq \mathcal{P}(x)f(y|x,\theta)$. It is defined as the expected value of the outer product of the score function $\nabla_\theta \log \mathcal{P}(x,y|\theta)$ with itself, evaluated at some $\theta \in \Omega$.

$$I_{\mathcal{P}(x)}(\theta) \triangleq \mathbb{E}_{(x,y)\sim\mathcal{P}(x,y|\theta)}[\nabla_\theta \log \mathcal{P}(x,y|\theta)\nabla_\theta^\mathsf{T} \log \mathcal{P}(x,y|\theta)]. \tag{6}$$

**Extended version of FI.** Here, we summarize the extended version of FI, which can be derived by making some assumptions. These variants of FI are utilized in the proof of Theorems.

- *(Hessian version)* Under the differentiability condition (A6) of Assumption 1 in Appendix F, FI can be written in terms of the Hessian matrix of the log-likelihood:

$$I_{\mathcal{P}(x)}(\theta) = -\mathbb{E}_{(x,y)\sim\mathcal{P}(x,y|\theta)}[\nabla_\theta^2 \log \mathcal{P}(x,y|\theta)]. \tag{7}$$

- *(Model decomposition version)* Under the factorization condition (A0) of Assumption 1 in Appendix F, (6) and (7) can be transformed as follows:

$$I_{\mathcal{P}(x)}(\theta) = \mathbb{E}_{(x,y)\sim\mathcal{P}(x)f(y|x,\theta)}[\nabla_\theta \log f(y|x,\theta)\nabla_\theta^\mathsf{T} \log f(y|x,\theta)] \tag{8}$$

$$= -\mathbb{E}_{(x,y)\sim\mathcal{P}(x)f(y|x,\theta)}[\nabla_\theta^2 \log f(y|x,\theta)]. \tag{9}$$

Specifically, (8) and (9) can be unfolded as follows:

$$I_{\mathcal{P}(x)}(\theta) = \int_{x\in X} \mathcal{P}(x)\sum_{y=1}^c \left[f(y|x,\theta)\cdot\nabla_\theta \log f(y|x,\theta)\nabla_\theta^\mathsf{T} \log f(y|x,\theta)\right]dx \tag{10}$$

$$= -\int_{x\in X} \mathcal{P}(x)\sum_{y=1}^c \left[f(y|x,\theta)\cdot\nabla_\theta^2 \log f(y|x,\theta)\right]dx$$

From now on, we define $I_{p(x)}(\theta)$ and $I_{q(x)}(\theta)$ as the FI derived from the training and test marginal, respectively.

### F.3.3 EMPIRICAL FISHER INFORMATION (EFI)

When the training dataset $\mathcal{D}_n$ is given, we denote the sampling probability as $h(x)$ which is defined on the probability space $\mathcal{H}$:

$$\mathcal{H} = \{h(x)\,|\,\textstyle\sum_{(x_i,y_i)\in\mathcal{D}_n} h(x_i) = 1\,,\,h(x_i) \geq 0 \quad \forall(x_i,y_i)\in\mathcal{D}_n\}.^{12} \tag{11}$$

Practically, the training dataset $\mathcal{D}_n$ is given as deterministic. Therefore, (8) can be refined as empirical Fisher information (EFI). This reformulation is frequently utilized, *e.g.,* in (Jastrzębski et al., 2017; Chaudhari et al., 2019), to reduce the computational complexity of gathering gradients for all possible classes (*i.e.,* expectation with respect to $f(y|x,\theta)$ as in (8)). Refer the $\sum_{y=1}^c$ term of (10). Different from prior EFI, which is defined on the case when $h(x)$ is uniform, $U(x)$, we generalize the definition of EFI in terms of $h(x) \in \mathcal{H}$ as follows:

$$\hat{I}_{h(x)}(\theta) := \mathbb{E}_{h(x)}\left[\nabla_\theta \log f(y|x,\theta)\nabla_\theta^\mathsf{T} \log f(y|x,\theta)\right]$$

$$\overset{(a)}{:=} \textstyle\sum_{(x_i,y_i)\in\mathcal{D}_n} h(x_i)\nabla_\theta \log f(y_i|x_i,\theta)\nabla_\theta^\mathsf{T} \log f(y_i|x_i,\theta). \tag{12}$$

Note that (a) holds owing to (11).

---

[12]Note that for simplicity, we abuse the notation $h(x,y)$ used in Section 3 as $h(x)$. This is exactly the same for a given dataset $\mathcal{D}_n$ situation.

### F.3.4 STOCHASTIC ORDER NOTATIONS $o_\mathrm{p}$ AND $O_\mathrm{p}$

For a set of random variables $X_n$ and a corresponding set of constant $a_n$, the notation $X_n = o_\mathrm{p}(a_n)$ means that the set of values $X_n/a_n$ converges to zero in probability as $n$ approaches an appropriate limit. It is equivalent with $X_n/a_n = o_\mathrm{p}(1)$, where $X_n = o_\mathrm{p}(1)$ is defined as:

$$\lim_{n\to\infty} \mathbb{P}(|X_n| \geq \epsilon) = 0 \quad \forall\,\epsilon \geq 0.$$

The notation $X_n = O_\mathrm{p}(a_n)$ means that the set of values $X_n/a_n$ is stochastically bounded. That is $\forall\,\epsilon > 0, \exists$ finite $M > 0,\ N > 0$ s.t. $\mathbb{P}(|X_n/a_n| > M) < \epsilon$ for any $n > N$.

## G  THEOREM 1

In this section, we deal with some required sub-lemmas that are used for the proof of Lemma 8, which is the main ingredient of the proof of Theorem 1.

### G.1  SUB-LEMMAS

**Lemma 1** ((Lehmann & Casella, 1998), Theorem 5.1). *When $\xrightarrow{P}$ denotes convergence in probability, and if (A0) to (A7) of the Assumption 1 in Appendix F hold, then there exists a sequence of MLE solutions $\{\hat{\theta}_{U(x),\mathcal{D}_n}\}_{n\in\mathbb{N}}$ that $\hat{\theta}_{U(x),\mathcal{D}_n} \xrightarrow{P} \theta^\star$ as $n \to \infty$, where $\theta^\star$ is the 'true' unknown parameter of the distribution of the sample.*

*Proof.* We refer to (Lehmann & Casella, 1998) for detailed proof. □

**Lemma 2** ((Lehmann & Casella, 1998), Theorem 5.1). *Let $\{\hat{\theta}_{U(x),\mathcal{D}_n}\}_{n\in\mathbb{N}}$ be the MLE based on the training data set $\mathcal{D}_n$. If (A0) to (A8) of the Assumption 1 in Appendix F hold, then the MLE $\hat{\theta}_{U(x),\mathcal{D}_n}$ has a zero-mean normal asymptotic distribution with a covariance equal to the inverse Fisher information matrix, and with the convergence rate of 1/2:*

$$\sqrt{n}(\hat{\theta}_{U(x),\mathcal{D}_n} - \theta^\star) \xrightarrow{D} \mathcal{N}(\vec{0}, I_{p(x)}(\theta^\star)^{-1}),$$

*where $\xrightarrow{D}$ represents convergence in distribution.*

*Proof.* We refer to (Lehmann & Casella, 1998) for detailed proof, based on Lemma 1. □

**Lemma 3** ((Wasserman, 2004), Theorem 9.18). *Under the (A0) to (A8) of the Assumption 1 in Appendix F hold, we get*

$$\sqrt{n}I_{p(x)}(\hat{\theta}_{U(x),\mathcal{D}_n})^{1/2}(\hat{\theta}_{U(x),\mathcal{D}_n} - \theta^\star) \xrightarrow{D} \mathcal{N}(\vec{0}, \mathbb{I}_d),$$

*where $\xrightarrow{D}$ represents convergence in distribution.*

*Proof.* We refer to (Wasserman, 2004) for detailed proof, based on Lemma 2. □

**Lemma 4.** *((Serfling, 1980), Chapter 1) Let $\{\theta_n\}$ be a sequence of random vectors. If there exists a random vector $\tilde{\theta}$ such that $\theta_n \xrightarrow{D} \tilde{\theta}$, then $\|\theta_n - \tilde{\theta}\|_2 = O_p(1)$, where $\|\cdot\|_2$ denote the $L_2$ norm.*

*Proof.* We refer to (Serfling, 1980) for detailed proof. □

**Lemma 5.** *((Sourati et al., 2016), Theorem 27) Let $\{\theta_n\}$ be a sequence of random vectors in a convex and compact set $\Omega \subseteq \mathbb{R}^d$ and $\theta^\star \in \Omega$ be a constant vector such that $\|\theta_n - \theta^\star\|_2 = O_p(a_n)$ where $a_n \to 0$ (as $n \to \infty$). If $g : \Omega \to \mathbb{R}$ is a $\mathcal{C}^3$ function, then*

$$g(\theta_n) = g(\theta^\star) + \nabla_\theta^\mathsf{T} g(\theta^\star)(\theta_n - \theta^\star) + \frac{1}{2}(\theta_n - \theta^\star)^\mathsf{T}\nabla_\theta^2 g(\theta^\star)(\theta_n - \theta^\star) + o_p(a_n^2).$$

*Proof.* We refer to (Serfling, 1980) for detailed proof. □

**Lemma 6.** *If (A0) and (A3) of the Assumption 1 in Appendix F hold, then $\nabla_\theta \log \mathcal{P}(x, y_{true}(x)|\theta^\star) = \vec{0}$ for any joint distribution $\mathcal{P}(x, y|\theta^\star)$.*

*Proof.*
$$\begin{aligned}
\nabla_\theta \log \mathcal{P}(x, y_{true}(x)|\theta^\star) &= \nabla_\theta \log f(y_{true}(x)|x, \theta^\star) \\
&= \nabla_\theta \log 1 \\
&= \vec{0}.
\end{aligned}$$

On the first equality, (A0) of Assumption 1 in Appendix F is used. At the second equality, (A3) of Assumption 1 in Appendix F is used.

□

**Lemma 7.** *If (A0) to (A8) of the Assumption 1 in Appendix F hold and the case $\nabla_\theta^2 \log p(x, y_{true}(x)|\theta^\star)$ is non-singular for given data $(x, y_{true}(x))$ satisfies, then the asymptotic distribution of the log-likelihood ratio is a mixture of first-order Chi-square distributions, and the convergence rate is one. More specifically:*

$$n \left( \log \frac{p(x, y_{true}(x)|\theta^\star)}{p(x, y_{true}(x)|\hat{\theta}_{U(x),\mathcal{D}_n})} \right) \xrightarrow{D} \frac{1}{2} \sum_{i=1}^d \lambda_i(x, y_{true}(x)) \cdot \chi_1^2, \tag{13}$$

*where $\{\lambda_i(x, y_{true}(x))\}_{i=1}^d$ are eigenvalues of $I_{p(x)}(\theta^\star)^{-\frac{1}{2}} \left\{ -\nabla_\theta^2 \log p(x, y_{true}(x)|\theta^\star) \right\} I_{p(x)}(\theta^\star)^{-\frac{1}{2}}$.*

*Proof.* The proof is based on the Taylor expansion theorem. Remind that we deal with the data $(x, y_{true}(x))$ satisfying $\nabla_\theta^2 \log p(x, y_{true}(x)|\theta^\star)$ is non-singular. From the property $\sqrt{n}(\hat{\theta}_{U(x),\mathcal{D}_n} - \theta^\star) \xrightarrow{D} \mathcal{N}(\vec{0}, I_{p(x)}(\theta^\star)^{-1})$ derived from Lemma 3, one concludes that $\sqrt{n}\|\hat{\theta}_{U(x),\mathcal{D}_n} - \theta^\star\|_2 = O_p(1)$ and therefore $\|\hat{\theta}_{U(x),\mathcal{D}_n} - \theta^\star\|_2 = O_p(\frac{1}{\sqrt{n}})$ by the Lemma 4.

Thus, by the Lemma 5,

$$\log p(x, y_{true}(x)|\hat{\theta}_{U(x),\mathcal{D}_n}) = \log p(x, y_{true}(x)|\theta^\star) + (\hat{\theta}_{U(x),\mathcal{D}_n} - \theta^\star)^\mathsf{T} \nabla_\theta \log p(x, y_{true}(x)|\theta^\star)$$

$$+ \frac{1}{2}(\hat{\theta}_{U(x),\mathcal{D}_n} - \theta^\star)^\mathsf{T} \nabla_\theta^2 \log p(x, y_{true}(x)|\theta^\star)(\hat{\theta}_{U(x),\mathcal{D}_n} - \theta^\star) + o_p\left(\frac{1}{n}\right)$$

holds. By the Lemma 3 and the property $\nabla_\theta \log p(x, y_{true}(x)|\theta^\star) = \vec{0}$ derived by Lemma 6, we can obtain

$$n \left[ \log p(x, y_{true}(x)|\theta^\star) - \log p(x, y_{true}(x)|\hat{\theta}_{U(x),\mathcal{D}_n}) \right]$$

$$= -\frac{1}{2} \left[ \sqrt{n}(\hat{\theta}_{U(x),\mathcal{D}_n} - \theta^\star) \right]^\mathsf{T} \nabla_\theta^2 \log p(x, y_{true}(x)|\theta^\star) \left[ \sqrt{n}(\hat{\theta}_{U(x),\mathcal{D}_n} - \theta^\star) \right] + o_p(1)$$

$$\xrightarrow{D} \frac{1}{2} \mathcal{N}\left(\vec{0}, I_{p(x)}(\theta^\star)^{-1}\right)^\mathsf{T} \left[ -\nabla_\theta^2 \log p(x, y_{true}(x)|\theta^\star) \right] \mathcal{N}\left(\vec{0}, I_{p(x)}(\theta^\star)^{-1}\right)$$

$$= \frac{1}{2} \mathcal{N}\left(\vec{0}, \mathbb{I}_d\right)^\mathsf{T} \left[ -I_{p(x)}(\theta^\star)^{-\frac{1}{2}} \nabla_\theta^2 \log p(x, y_{true}(x)|\theta^\star) I_{p(x)}(\theta^\star)^{-\frac{1}{2}} \right] \mathcal{N}\left(\vec{0}, \mathbb{I}_d\right).$$

Define $\Gamma(x, y_{true}(x))$ as $-I_{p(x)}(\theta^\star)^{-\frac{1}{2}} \nabla_\theta^2 \log p(x, y_{true}(x)|\theta^\star) I_{p(x)}(\theta^\star)^{-\frac{1}{2}}$ and rewrite the right-hand-side element-wise[13] as

$$\frac{1}{2} \mathcal{N}\left(\vec{0}, \mathbb{I}_d\right)^\mathsf{T} \Gamma(x, y_{true}(x)) \mathcal{N}\left(\vec{0}, \mathbb{I}_d\right)$$

$$= \frac{1}{2} \sum_{i=1}^d \lambda_i(x, y_{true}(x)) \cdot \mathcal{N}(0, 1)^2 = \frac{1}{2} \sum_{i=1}^d \lambda_i(x, y_{true}(x)) \cdot \chi_1^2,$$

where $\{\lambda_i(x, y_{true}(x))\}_{i=1}^d$ are eigenvalues of $\Gamma(x, y_{true}(x))$. Thus, the desired property is obtained. $\square$

### G.2 MAIN LEMMA

In this section, we derive the main Lemma, which represents the test cross-entropy loss and can be understood as Fisher information ratio (FIR) (Sourati et al., 2016).

---

[13]Suppose $\Gamma = U\Sigma U^\mathsf{T}$ and $\Sigma = diag(\lambda_1, \cdots \lambda_d)$. Then, $\mathcal{N}\left(\vec{0}, \mathbb{I}_d\right)^\mathsf{T} U \sim \mathcal{N}\left(\vec{0}, UU^\mathsf{T}\right) = \mathcal{N}\left(\vec{0}, \mathbb{I}_d\right)$. Thus, $\mathcal{N}\left(\vec{0}, \mathbb{I}_d\right)^\mathsf{T} \Gamma \mathcal{N}\left(\vec{0}, \mathbb{I}_d\right) = \mathcal{N}\left(\vec{0}, \mathbb{I}_d\right)^\mathsf{T} \Sigma \mathcal{N}\left(\vec{0}, \mathbb{I}_d\right) = \sum_{i=1}^d \lambda_i \mathcal{N}(0, 1)^2$.

### G.2.1 MAIN LEMMA STATEMENT AND PROOF

**Lemma 8** (FIR in expected test cross entropy loss with MLE). *If the Assumption 1 in Appendix F holds, then*

$$\lim_{n\to\infty} n\mathbb{E}_{(x,y)\sim q(x)f(y|x,\theta^\star)} \left[ \mathbb{E}_{\mathcal{D}_n\sim p(x)f(y|x,\theta^\star)} \left[ -\log f(y|x,\hat{\theta}_{U(x),\mathcal{D}_n}) \right] \right]$$
$$= \frac{1}{2}\operatorname{Tr}\left[ I_{p(x)}(\theta^\star)^{-1} I_{q(x)}(\theta^\star) \right]. \tag{14}$$

*Proof.* We prove Lemma 8 via two steps. First we show that the expected cross-entropy loss term can be rewritten in terms of the log-likelihood ratio. Then, we prove that the expected log-likelihood ratio can be asymptotically understood as FIR.

**Step 1: Log-likelihood ratio** We show that the expected log-likelihood ratio can be formulated as the expected test cross-entropy loss as follows:

This property holds because,

$$\mathbb{E}_{(x,y)\sim q(x)f(y|x,\theta^\star)} \left[ \mathbb{E}_{\mathcal{D}_n\sim p(x)f(y|x,\theta^\star)} \left[ \log \frac{p(x,y|\theta^\star)}{p(x,y|\hat{\theta}_{U(x),\mathcal{D}_n})} \right] \right]$$

$$= \mathbb{E}_{(x,y)\sim q(x)f(y|x,\theta^\star)} \left[ \mathbb{E}_{\mathcal{D}_n\sim p(x)f(y|x,\theta^\star)} \left[ \log \frac{f(y|x,\theta^\star)}{f(y|x,\hat{\theta}_{U(x),\mathcal{D}_n})} \right] \right] \tag{15}$$

$$= \mathbb{E}_{(x,y)\sim q(x)f(y|x,\theta^\star)} \left[ \mathbb{E}_{\mathcal{D}_n\sim p(x)f(y|x,\theta^\star)} \left[ \log \frac{f(y|x,\theta^\star)}{f(y|x,\hat{\theta}_{U(x),\mathcal{D}_n})} \right] \mathbb{1}_{Supp(q(x,y|\theta^\star))} \right] \tag{16}$$

$$= \mathbb{E}_{(x,y)\sim q(x)f(y|x,\theta^\star)} \left[ \mathbb{E}_{\mathcal{D}_n\sim p(x)f(y|x,\theta^\star)} \left[ \log \frac{f(y|x,\theta^\star)}{f(y|x,\hat{\theta}_{U(x),\mathcal{D}_n})} \mathbb{1}_{Supp(q(x,y|\theta^\star))} \right] \right]$$

$$= \mathbb{E}_{(x,y)\sim q(x)f(y|x,\theta^\star)} \left[ \mathbb{E}_{\mathcal{D}_n\sim p(x)f(y|x,\theta^\star)} \left[ -\log f(y|x,\hat{\theta}_{U(x),\mathcal{D}_n}) \mathbb{1}_{Supp(q(x,y|\theta^\star))} \right] \right] \tag{17}$$

$$= \mathbb{E}_{(x,y)\sim q(x)f(y|x,\theta^\star)} \left[ \mathbb{E}_{\mathcal{D}_n\sim p(x)f(y|x,\theta^\star)} \left[ -\log f(y|x,\hat{\theta}_{U(x),\mathcal{D}_n}) \right] \right]$$

Since (A0) of Assumption 1 in Appendix F, (15) and (16) hold. At (17), the properties, (i) $Supp(q(x,y|\theta^\star)) \subseteq \mathcal{W}$ and (ii) $f(y|x,\theta^\star) = 1 \ \forall (x,y) \in \mathcal{W}$, was used which is derived by (A3) of the Assumption 1 in Appendix F.

**Step 2: FIR** Here, we show that the expected test loss of MLE can be understood as FIR.

By (A0) in Assumption 1 in Appendix F, trivially

$$\{(x,y) \in Supp(q(x,y|\theta^\star)) \mid \nabla_\theta^2 \log q(x,y|\theta^\star) \text{ is singular}\}$$
$$= \{(x,y) \in Supp(q(x,y|\theta^\star)) \mid \nabla_\theta^2 \log p(x,y|\theta^\star) \text{ is singular}\}. \tag{18}$$

holds. Since (18), and (A9) in Assumption 1 in Appendix F, $Supp(q(x,y|\theta^\star))$ can be replaced by,

$$S \triangleq Supp(q(x,y|\theta^\star)) \setminus \{(x,y) \in \mathcal{W} \mid \nabla_\theta^2 \log p(x,y|\theta^\star) \text{ is singular}\} \tag{19}$$

when calculate expectation.

We can get a result of Lemma 8 as follows:

$$\lim_{n\to\infty} n\mathbb{E}_{(x,y)\sim q(x)f(y|x,\theta^\star)} \left[ \mathbb{E}_{\mathcal{D}_n\sim p(x)f(y|x,\theta^\star)} \left[ -\log f(y|x,\hat{\theta}_{U(x),\mathcal{D}_n}) \right] \right]$$

$$= \lim_{n\to\infty} n\mathbb{E}_{(x,y)\sim q(x)f(y|x,\theta^\star)} \left[ \mathbb{E}_{\mathcal{D}_n\sim p(x)f(y|x,\theta^\star)} \left[ \log \frac{p(x,y|\theta^\star)}{p(x,y|\hat{\theta}_{U(x),\mathcal{D}_n})} \right] \mathbf{1}_{Supp(q(x,y|\theta^\star))} \right] \tag{20}$$

$$= \lim_{n\to\infty} n\mathbb{E}_{(x,y)\sim q(x)f(y|x,\theta^\star)} \left[ \mathbb{E}_{\mathcal{D}_n\sim p(x)f(y|x,\theta^\star)} \left[ \log \frac{p(x,y|\theta^\star)}{p(x,y|\hat{\theta}_{U(x),\mathcal{D}_n})} \right] \mathbf{1}_S \right] \tag{21}$$

$$= \mathbb{E}_{(x,y)\sim q(x)f(y|x,\theta^\star)} \left[ \lim_{n\to\infty} \mathbb{E}_{\mathcal{D}_n\sim p(x)f(y|x,\theta^\star)} \left[ n\log \frac{p(x,y|\theta^\star)}{p(x,y|\hat{\theta}_{U(x),\mathcal{D}_n})} \right] \mathbf{1}_S \right]$$

$$= \mathbb{E}_{(x,y_{true}(x)) \sim q(x)f(y|x,\theta^\star)} \left[ \lim_{n \to \infty} \mathbb{E}_{\mathcal{D}_n \sim p(x)f(y|x,\theta^\star)} \left[ n \log \frac{p(x, y_{true}(x)|\theta^\star)}{p(x, y_{true}(x)|\hat{\theta}_{U(x),\mathcal{D}_n})} \right] \mathbf{1}_S \right] \tag{22}$$

$$= \mathbb{E}_{(x,y_{true}(x)) \sim q(x)f(y|x,\theta^\star)} \left[ \left( \frac{1}{2} \sum_{i=1}^d \lambda_i(x, y_{true}(x)) \mathbb{E}\left[\chi_1^2\right] \right) \mathbf{1}_S \right] \tag{23}$$

$$= \mathbb{E}_{(x,y_{true}(x)) \sim q(x)f(y|x,\theta^\star)} \left[ \left( \frac{1}{2} \sum_{i=1}^d \lambda_i(x, y_{true}(x)) \right) \mathbf{1}_S \right] \tag{24}$$

$$= \mathbb{E}_{(x,y_{true}(x)) \sim q(x)f(y|x,\theta^\star)} \left[ \frac{1}{2} \operatorname{Tr}\left[ I_{p(x)}(\theta^\star)^{-\frac{1}{2}} \left\{-\nabla_\theta^2 \log p(x, y_{true}(x)|\theta^\star)\right\} I_{p(x)}(\theta^\star)^{-\frac{1}{2}} \right] \mathbf{1}_S \right] \tag{25}$$

$$= \mathbb{E}_{(x,y) \sim q(x)f(y|x,\theta^\star)} \left[ \frac{1}{2} \operatorname{Tr}\left[ I_{p(x)}(\theta^\star)^{-\frac{1}{2}} \left\{-\nabla_\theta^2 \log p(x, y|\theta^\star)\right\} I_{p(x)}(\theta^\star)^{-\frac{1}{2}} \right] \mathbf{1}_S \right] \tag{26}$$

$$= \frac{1}{2} \operatorname{Tr}\left[ I_{p(x)}(\theta^\star)^{-\frac{1}{2}} \mathbb{E}_{(x,y) \sim q(x)f(y|x,\theta^\star)} \left[ -\nabla_\theta^2 \log p(x, y|\theta^\star) \, \mathbf{1}_S \right] I_{p(x)}(\theta^\star)^{-\frac{1}{2}} \right]$$

$$= \frac{1}{2} \operatorname{Tr}\left[ I_{p(x)}(\theta^\star)^{-\frac{1}{2}} \mathbb{E}_{(x,y) \sim q(x)f(y|x,\theta^\star)} \left[ -\nabla_\theta^2 \log q(x, y|\theta^\star) \, \mathbf{1}_S \right] I_{p(x)}(\theta^\star)^{-\frac{1}{2}} \right] \tag{27}$$

$$= \frac{1}{2} \operatorname{Tr}\left[ I_{p(x)}(\theta^\star)^{-\frac{1}{2}} \mathbb{E}_{(x,y) \sim q(x)f(y|x,\theta^\star)} \left[ -\nabla_\theta^2 \log q(x, y|\theta^\star) \right] I_{p(x)}(\theta^\star)^{-\frac{1}{2}} \right]$$

$$= \frac{1}{2} \operatorname{Tr}\left[ I_{p(x)}(\theta^\star)^{-\frac{1}{2}} I_{q(x)}(\theta^\star) I_{p(x)}(\theta^\star)^{-\frac{1}{2}} \right]$$

$$= \frac{1}{2} \operatorname{Tr}\left[ I_{p(x)}(\theta^\star)^{-1} I_{q(x)}(\theta^\star) \right]. \tag{28}$$

(20) holds from Step 1. From (19), (21) satisfied. (22) and (26) holds because $(x, y)$ is sampled from $q(x)f(y|x.\theta^\star)$. From (23) to (25), the result of Lemma 7 is used. (27) can be obtained thanks to (A0). Lastly, (28) holds because trace satisfies the commutative law about matrix multiplication.

The final term, $\operatorname{Tr}\left[I_{p(x)}(\theta^\star)^{-1} I_{q(x)}(\theta^\star)\right]$, is known as the Fisher information ratio (FIR) because it can be expressed as a ratio in the scalar case.

$\square$

### G.3 THEOREM 1

In this section, we finally prove Theorem 1. To do so, we additionally follow assumptions in (Sourati et al., 2016).

#### G.3.1 ADDITIONAL ASSUMPTION

**Assumption 2.** We assume to exist four positive constants $L_1, L_2, L_3, L_4 \geq 0$ such that following properties hold $\forall x \in X, y \in \{1, \cdots, c\}$ and $\theta \in \Omega$ :

- $I(\theta, x) = -\nabla_\theta^2 \log f(y|x, \theta)$ is independent of the class labels $y$.

- $\nabla_\theta \log f(y|x, \theta^\star)^\mathsf{T} I_{q(x)}(\theta^\star)^{-1} \nabla_\theta \log f(y|x, \theta^\star) \leq L_1$

- $\|I_{q(x)}(\theta^\star)^{-1/2} I(\theta^\star, x) I_{q(x)}(\theta^\star)^{-1/2}\| \leq L_2$

- $\|I_{q(x)}(\theta^\star)^{-1/2}(I(\theta', x) - I(\theta'', x)) I_{q(x)}(\theta^\star)^{-1/2}\| \leq L_3(\theta' - \theta'')^\mathsf{T} I_{q(x)}(\theta^\star)(\theta' - \theta'')$

- $-L_4\|\theta - \theta^\star\| I(\theta^\star, x) \preceq I(\theta, x) - I(\theta^\star, x) \preceq L_4\|\theta - \theta^\star\| I(\theta^\star, x)$

### G.3.2 REPLACING $\theta^\star$ BY $\hat{\theta}_{U(x),\mathcal{D}_n}$

**Lemma 9.** *Suppose Assumption 1 in Appendix F and Assumption 2 in Appendix G hold, then with high probability:*

$$\mathrm{Tr}\left[I_{p(x)}(\theta^\star)^{-1}\right] = \lim_{n\to\infty} \mathrm{Tr}\left[I_{p(x)}(\hat{\theta}_{U(x),\mathcal{D}_n})^{-1}\right]. \tag{29}$$

*Proof.* It is shown in the proof of Lemma 2 in (Chaudhuri et al., 2015) that under the assumptions mentioned in Assumption 2, the following inequalities hold with probability $1 - \delta(n)$:

$$\frac{\beta(n)-1}{\beta(n)} I(\theta^\star, x) \preceq I(\hat{\theta}_{U(x),\mathcal{D}_n}, x) \preceq \frac{\beta(n)+1}{\beta(n)} I(\theta^\star, x), \tag{30}$$

where $\beta(n)$ and $1 - \delta(n)$ are proportional to $n$, which is the size of the training set $\mathcal{D}_n$.

Because of the independence for the class labels $y$ of $I(\theta, x)$, $\mathcal{P}(x)$, $I_{\mathcal{P}(x)}(\theta) = \mathbb{E}_{x\sim\mathcal{P}(x)}\left[I(\theta, x)\right]$ holds for any marginal distribution $\mathcal{P}(x)$.[14]

Taking the expectation to the (30) with respect to the marginal $p(x)$ and $q(x)$, then:

$$\frac{\beta(n)-1}{\beta(n)} I_{p(x)}(\theta^\star) \preceq I_{p(x)}(\hat{\theta}_{U(x),\mathcal{D}_n}) \preceq \frac{\beta(n)+1}{\beta(n)} I_{p(x)}(\theta^\star). \tag{31}$$

$$\frac{\beta(n)-1}{\beta(n)} I_{q(x)}(\theta^\star) \preceq I_{q(x)}(\hat{\theta}_{U(x),\mathcal{D}_n}) \preceq \frac{\beta(n)+1}{\beta(n)} I_{q(x)}(\theta^\star).$$

Since $I_{p(x)}(\theta^\star)$ and $I_{p(x)}(\hat{\theta}_{U(x),\mathcal{D}_n})$ are assumed to be positive definite, we can write (31) in terms of inverted matrices[15]:

$$\frac{\beta(n)}{\beta(n)+1} I_{p(x)}(\theta^\star)^{-1} \preceq I_{p(x)}(\hat{\theta}_{U(x),\mathcal{D}_n})^{-1} \preceq \frac{\beta(n)}{\beta(n)-1} I_{p(x)}(\theta^\star)^{-1}. \tag{32}$$

(32) is equivalent to

$$\frac{\beta(n)-1}{\beta(n)} I_{p(x)}(\hat{\theta}_{U(x),\mathcal{D}_n})^{-1} \preceq I_{p(x)}(\theta^\star)^{-1} \preceq \frac{\beta(n)+1}{\beta(n)} I_{p(x)}(\hat{\theta}_{U(x),\mathcal{D}_n})^{-1}. \tag{33}$$

From (33),

$$\frac{\beta(n)-1}{\beta(n)} \mathrm{Tr}\left[I_{p(x)}(\hat{\theta}_{U(x),\mathcal{D}_n})^{-1}\right] \leq \mathrm{Tr}\left[I_{p(x)}(\theta^\star)^{-1}\right] \leq \frac{\beta(n)+1}{\beta(n)} \mathrm{Tr}\left[I_{p(x)}(\hat{\theta}_{U(x),\mathcal{D}_n})^{-1}\right] \tag{34}$$

satisfies.[16]

Thus,

$$\lim_{n\to\infty} \mathrm{Tr}\left[I_{p(x)}(\hat{\theta}_{U(x),\mathcal{D}_n})^{-1}\right] = \mathrm{Tr}\left[I_{p(x)}(\theta^\star)^{-1}\right] \tag{35}$$

holds when taking $n \to \infty$ to the (34). Note that $\beta(n)$ is proportional to $n$.

$\square$

---

[14] $I_{\mathcal{P}(x)}(\theta) = \mathbb{E}_{(x,y)\sim\mathcal{P}(x,y|\theta)}\left[-\nabla_\theta^2 \log f(y|x,\theta)\right] = \mathbb{E}_{x\sim\mathcal{P}(x)}\left[\mathbb{E}_{y\sim f(y|x,\theta)}\left[-\nabla_\theta^2 \log f(y|x,\theta)\right]\right] = \mathbb{E}_{x\sim\mathcal{P}(x)}\left[\mathbb{E}_{y\sim f(y|x,\theta)}\left[I(\theta, x)\right]\right] = \mathbb{E}_{x\sim\mathcal{P}(x)}\left[I(\theta, x)\right]$.

[15] For $\forall$ two positive definite matrices $A$ and $B$, we have that $A \succeq B \Rightarrow A^{-1} \preceq B^{-1}$

[16] If $A \preceq B$, $\mathrm{Tr}[A] \leq \mathrm{Tr}[B]$ holds.
($\because$) $A \preceq B \Rightarrow B - A \succeq \mathcal{O}$ and $B - A := U\Sigma U^T$, where $U = [u_1|\cdots,|u_d]$.
Then, $\mathrm{Tr}(B - A) = \sum_{i=1}^d u_i(B - A)u_i^T \leq 0$ because of the positive definite property of $B - A$.
$\mathrm{Tr}(B - A) \leq 0 \Rightarrow \mathrm{Tr}(B) \leq \mathrm{Tr}(A)$.

## G.4 Statement and proof of Theorem 1

**Theorem 1.** *Suppose Assumption 1 in Appendix F and Assumption 2 in Appendix G hold, then for sufficiently large $n = |\mathcal{D}_n|$, the following holds with high probability:*

$$
\begin{aligned}
\mathbb{E}_{(x,y)\sim q(x)f(y|x,\theta^\star)} &\left[ \mathbb{E}_{\mathcal{D}_n \sim p(x)f(y|x,\theta^\star)} \left[ -\log f(y|x,\hat{\theta}_{U(x),\mathcal{D}_n}) \right] \right] \\
&\leq \frac{1}{2n} \operatorname{Tr}\left[ I_{p(x)}(\hat{\theta}_{U(x),\mathcal{D}_n})^{-1} \right] \operatorname{Tr}\left[ I_{q(x)}(\theta^\star) \right].
\end{aligned}
\tag{36}
$$

*Proof.* Because of the (A8) of Assumption 1 in Appendix F, $I_{p(x)}(\theta^\star)^{-1}$ and $I_{q(x)}(\theta^\star)$ are positive definite matrix. Thus, $\operatorname{Tr}\left[ I_{p(x)}(\theta^\star)^{-1} I_{q(x)}(\theta^\star) \right] \leq \operatorname{Tr}\left[ I_{p(x)}(\theta^\star)^{-1} \right] \operatorname{Tr}\left[ I_{q(x)}(\theta^\star) \right]$ holds.[17]

From the result of Lemma 8 and 9 in Appendix G,

$$
\begin{aligned}
\lim_{n\to\infty} n\mathbb{E}_{(x,y)\sim q(x)f(y|x,\theta^\star)} &\left[ \mathbb{E}_{\mathcal{D}_n \sim p(x)f(y|x,\theta^\star)} \left[ -\log f(y|x,\hat{\theta}_{U(x),\mathcal{D}_n}) \right] \right] \\
&= \frac{1}{2} \operatorname{Tr}\left[ I_{p(x)}(\theta^\star)^{-1} I_{q(x)}(\theta^\star) \right] \\
&\leq \frac{1}{2} \operatorname{Tr}\left[ I_{p(x)}(\theta^\star)^{-1} \right] \operatorname{Tr}\left[ I_{q(x)}(\theta^\star) \right] \\
&= \frac{1}{2} \lim_{n\to\infty} \operatorname{Tr}\left[ I_{p(x)}(\hat{\theta}_{U(x),\mathcal{D}_n})^{-1} \right] \operatorname{Tr}\left[ I_{q(x)}(\theta^\star) \right]
\end{aligned}
$$

holds with high probability. $\qquad\square$

It is worth noting that Theorem 1 states that the upper bound of the MLE $\hat{\theta}_{U(x),\mathcal{D}_n}$, $\mathcal{D}_n$ test loss can be minimized by lowering $\operatorname{Tr}\left[ I_{p(x)}(\hat{\theta}_{U(x),\mathcal{D}_n})^{-1} \right]$ when training marginal $p(x)$, the only tractable and controllable variable.

## H Theorem 2

In this section, we introduce the motivation of gradient norm-based importance sampling. To show why this is important, we introduce the debiasing object problem for a given $\mathcal{D}_n$ under sampling probability $h(x)$ and show how to solve it in a toy example because the problem is difficult.

### H.1 Practical objective function for the dataset bias problem

Remark that the right-hand side term of (36) are controlled by the training and test marginals $p(x)$, and $q(x)$. Since we can only control the training dataset $\mathcal{D}_n$ not $p(x)$ and $q(x)$, we can design a practical objective function for the dataset bias problem by using EFI and Theorem 1 as follows:

$$
\min_{h(x)\in\mathcal{H}} \operatorname{Tr}\left[ \hat{I}_{h(x)}(\hat{\theta}_{h(x),\mathcal{D}_n})^{-1} \right],
\tag{37}
$$

where $\hat{I}_{h(x)}(\theta)$ is an empirical Fisher information matrix. Remark that EFI is defined as:

$$
\hat{I}_{h(x)}(\theta) = \sum_{i=1}^{n} h(x_i)\nabla_\theta \log f(y_i|x_i,\theta)\nabla_\theta^\top \log f(y_i|x_i,\theta).
\tag{38}
$$

Here, $h(x)$ describes the sampling probability on $\mathcal{D}_n$, which is the only controllable term. We deal with (37) in the toy example because of the difficulty of the problem.

### H.2 One-dimensional Toy Example Setting

For simplicity, we assume that $\mathcal{D}_n$ comprises sets $M$ and $m$, and the samples in each set share the same loss function and the same probability mass. The details are as follows:

---

[17]For $\forall$ two positive definite matrices $A$ and $B$, $\operatorname{Tr}[AB] \leq \operatorname{Tr}[A]\operatorname{Tr}[B]$ satisfies.

- For the given $a \in \mathbb{R}$, at the model parameter $\theta \in \mathbb{R}$, $\frac{1}{2}(\theta + a)^2$ and $\frac{1}{2}(\theta - a)^2$ loss function arise for all data in $M$ and $m$, respectively.

- $\hat{\theta}_{h, \mathcal{D}_n}$ denote the trained model from the arbitrary PMF $h(x) \in \mathcal{H}$ which has a constraint having degree of freedom 2, $(h_M(x), h_m(x))$.

- Concretely, each samples of $M$ and $m$ has a probability mass $h_M(x)$ and $h_m(x)$, respectively. *i.e.*, $|M| \cdot h_M(x) + |m| \cdot h_m(x) = 1$, where $|M|$ and $|m|$ denote the cardinality of $M$ and $m$, respectively.

- Let $g_M(\theta)$ and $g_m(\theta)$ denote the gradient of each sample in $M$ and $m$ at $\theta \in \mathbb{R}$, respectively.

- Then, $|M| \cdot h_M(x) \cdot g_M(\hat{\theta}_{h(x), \mathcal{D}_n}) + |m| \cdot h_m(x) \cdot g_m(\hat{\theta}_{h(x), \mathcal{D}_n}) = 0$ hold by the definition of $\hat{\theta}_{h(x), \mathcal{D}_n}$.

- In these settings, our objective can be written as finding $h^\star(x) = \arg\min_{h(x) \in \mathcal{H}} \mathrm{Tr}\left[\hat{I}_h(x)(\hat{\theta}_{h(x), \mathcal{D}_n})^{-1}\right]$ and this is equivalent to find $(h_M^\star(x), h_m^\star(x))$.

## H.3    Statement and proof of Theorem 2

In this section, we introduce the motivation for the gradient norm-based importance sampling in the toy example setting.

**Theorem 2.** *Under the above setting, the solution of* $(h_M^\star(x), h_m^\star(x)) = \arg\min_{h(x) \in \mathcal{H}} \mathrm{Tr}\left[\hat{I}_{h(x)}(\hat{\theta}_{h(x), \mathcal{D}_n})^{-1}\right]$ *is:*

$$h_M^\star(x) = \frac{|g_M(\hat{\theta}_{U(x), \mathcal{D}_n})|}{Z}, \quad h_m^\star(x) = \frac{|g_m(\hat{\theta}_{U(x), \mathcal{D}_n})|}{Z},$$

*where* $Z = |M||g_M(\hat{\theta}_{U(x), \mathcal{D}_n})| + |m||g_m(\hat{\theta}_{U(x), \mathcal{D}_n})|$, *and* $|M|$ *and* $|m|$ *denote the cardinality of* $M$ *and* $m$, *respectively.*

*Proof.* The trained model $\hat{\theta}_{h(x), \mathcal{D}_n} \in [-a, a]$ holds trivially for any $h(x) \in \mathcal{H}$. By the loss function definition in the toy setting, $g_M(\theta) = \theta + a$ and $g_m(\theta) = \theta - a$, $\forall \theta$. Thus, $|g_M(\theta)| + |g_m(\theta)| = 2a$ satisfies for $\forall \theta \in [-a, a]$. Since the gradient is scalar in the toy setting, $\hat{I}_{h(x)}(\hat{\theta}_{h(x), \mathcal{D}_n})$ is also scalar and the same as the unique eigenvalue, that is,

$$\hat{I}_{h(x)}(\hat{\theta}_{h(x), \mathcal{D}_n}) = |M| \cdot h_M(x) \cdot \{g_M(\hat{\theta}_{h(x), \mathcal{D}_n})\}^2 + |m| \cdot h_m(x) \cdot \{g_m(\hat{\theta}_{h(x), \mathcal{D}_n})\}^2 \quad \forall h(x) \in \mathcal{H}.$$

Thus, our problem is deciding $h_M(x)$ and $h_m(x)$ that maximize $|M| \cdot h_M(x) \cdot \{g_M(\hat{\theta}_{h(x), \mathcal{D}_n})\}^2 + |m| \cdot h_m(x) \cdot \{g_m(\hat{\theta}_{h(x), \mathcal{D}_n})\}^2$.
Because of the toy setting, three constraints are held for arbitrary $\theta \in [-a, a]$ and $h(x) \in \mathcal{H}$.

1. $|M| \cdot h_M(x) + |m| \cdot h_m(x) = 1$. (probability definition)

2. $|M| \cdot h_M(x) \cdot g_M(\hat{\theta}_{h(x), \mathcal{D}_n}) + |m| \cdot h_m(x) \cdot g_m(\hat{\theta}_{h(x), \mathcal{D}_n}) = 0$.
   Note that convex linear sum of the sample's gradient w.r.t. $h(x) = (h_M(x), h_m(x))$ is zero at the trained model $\hat{\theta}_{h(x), \mathcal{D}_n}$.

3. $|g_M(\theta)| + |g_m(\theta)| = 2a \Leftrightarrow g_M(\theta) - g_m(\theta) = 2a \Leftrightarrow g_M(\theta) = 2a + g_m(\theta)$.
   Note that this is derived by the property of predefined loss function at $\theta \in [-a, a]$.

2nd constraint is equivalent to $|M| \cdot h_M(x) \cdot (2a + g_m(\hat{\theta}_{h(x), \mathcal{D}_n})) + |m| \cdot h_m(x) \cdot g_m(\hat{\theta}_{h(x), \mathcal{D}_n}) = 0$. $\Leftrightarrow g_m(\hat{\theta}_{h(x), \mathcal{D}_n}) = -2a|M| \cdot h_M(x)$. Because of the 3rd constraint, $g_M(\hat{\theta}_{h(x), \mathcal{D}_n}) = 2a(1 - |M| \cdot h_M(x))$. Then the objective is, maximizing

$$|M| \cdot h_M(x) \cdot \{2a(1 - |M| \cdot h_M(x))\}^2 + (1 - |M| \cdot h_M(x))\{2a|M| \cdot h_M(x)\}^2 \tag{39}$$
$$= 4a^2 |M| \cdot h_M(x)(1 - |M| \cdot h_M(x))$$

(39) is maximized when $|M| \cdot h_M(x) = \frac{1}{2}$, and it means $|m| \cdot h_m(x) = \frac{1}{2}$. Thus, $h_M^\star(x) = $

$\frac{|m|}{2|M|\cdot|m|}$, $h_m^\star(x) = \frac{|M|}{2|M|\cdot|m|}$. This result is related with the trained model $\hat{\theta}_{U(x),\mathcal{D}_n}$, where $U_M(x) = U_m(x) = \frac{1}{|M|+|m|}$. At $\hat{\theta}_{U(x),\mathcal{D}_n}$, $|M|g_M(\hat{\theta}_{U(x),\mathcal{D}_n}) + |m|g_m(\hat{\theta}_{U(x),\mathcal{D}_n}) = 0$ satisfies and this is equivalent to $|M| : |m| = |g_m(\hat{\theta}_{U(x),\mathcal{D}_n})| : |g_M(\hat{\theta}_{U(x),\mathcal{D}_n})|$. □

Thus, it is consistent with our intuition that setting the sampling probability $h$ for set M and m in proportion to $|g_M(\hat{\theta}_{U(x),\mathcal{D}_n})|$ and $|g_m(\hat{\theta}_{U(x),\mathcal{D}_n})|$ helps to minimize the trace of the inverse empirical Fisher information.

