# OpenReview forum: "Mitigating Dataset Bias by Using Per-Sample Gradient"
_ICLR.cc/2023/Conference — ICLR 2023 poster_

### Official Review · Reviewer_Lo2Z · 2022-10-23

**Confidence:** 3
**Correctness:** 4
**Technical Novelty And Significance:** 4
**Empirical Novelty And Significance:** 4
**Recommendation:** 8

**Clarity, Quality, Novelty And Reproducibility:**

The paper is clearly written with solid theoretical justification and extensive experiments.




**Strength And Weaknesses:**

Strength:
* Comparing to previous approach, this current paper proposed a distinct signal (gradient magnitude) for quantifying the learning difficulty of examples, and proposed a simple learning-time resampling approach to debiasing.
* The proposed methodology is simple and general purpose, and therefore worthwhile put it out there for the community.
* The method is comprehensively evaluated across multiple benchmarks and comes with sufficient ablation study.

Weakness:
* None.

**Summary Of The Paper:**

This work presents a interesting approach to achieve group robustness without group label by re-sampling training data based on last-layer gradients, the work is simple, comes with rigorous theoretical justification and extensive experiments.

**Summary Of The Review:**

As stated in the summary, I find the work to be simple, comes with rigorous theoretical justification and extensive experiments. Therefore I believe it carries sufficient scientific novelty and empirical significance to justify an acceptance.

---

> ### Author Response · Authors · 2022-11-12
> **Response to Reviewer Lo2Z**
>
> We are encouraged that the reviewer has pointed out the strength of how the proposed method measures the learning difficulty by using gradient information (and thus solve dataset bias), rather than loss or features space, which have been frequently utilized. In addition, the reviewer valued the simplicity and applicability of the method. Finally, our justification through various experiments, theories, and empirical analysis are highly valued. We are very grateful for this assesment.

---

### Official Review · Reviewer_oD49 · 2022-10-24

**Confidence:** 4
**Correctness:** 3
**Technical Novelty And Significance:** 3
**Empirical Novelty And Significance:** 3
**Recommendation:** 8

**Clarity, Quality, Novelty And Reproducibility:**

**Clarity.** While the idea of the paper is rather intuitive and easy to follow, the writing needs a full revision. A lot of phrases are rather confusing, or directly grammatically incorrect (e.g., "PGD outperforms dataset bias", "Almost hair color of female images are blond"). More on this note, I could not understand the 1D toy, as I couldn't parse the phrase "elements in each set share the same loss function".

Regarding clarity, section 5 and the appendix are specially difficult to parse, with heavy maths that contrast the rest of the paper. I am confident it could be written in a more understandable way.

**Quality.** I think quaility is ok, disregarding the rest of points I've made. _I have not checked the proofs._

**Novelty.** While the method is novel, I do believe it used LfF a lot for inspiration, which might be worth mentioning. This is specially clear when trying to understand design choices such as using GCE for the first model, but then evaluating in CE. To understand the reasoning behind these choices, I had to go back to the LfF paper and re-read it.

Regarding connections, I find the objective in Eq. (6) to be quite similar to that of [non-probabilistic robust optimization](https://en.wikipedia.org/wiki/Robust_optimization#Probabilistically_robust_optimization_models), so it might be worth mentioning.

**Reproduciblity.** I have not checked the code myself, but it should be reproducibe. On this note, differences with the results from the LfF paper should be clarified.

**Strength And Weaknesses:**

**Strengths:**
- S1. The algorithm is clear, simple, effective, and therefore of interest to the community.
- S2. Experimental results are strong, and different ablation studies were performed.
- S3. Theoretical justification is appealing and sensible.

**Weaknesses:**
- W1. Different ablation studies would be needed to understand where does the improvement come from. Specifically, the proposed approach is clearly inspired by LfF, and the difference are: i) sequential training of both models, instead of interleaved steps; ii) different weighting function; iii) re-sampling instead of re-weighting gradients (+ data augmentation). A proper ablation study should shed some lights on which steps are giving an edge to the proposed approach with respect to LfF.
   - **Edit.** I do not want to remove the point, as I think it is still important to clarify in the main paper, but I have just read Appendix C6 and C7 where part of these studies are done. After reading them, it is unclear to me whether using the gradient norm improves the results, as the proposed method with single-stage and reweighting performs similar to LfF.
- W2. I am not sure how fair it is to introduce the data augmentation as part of the approach. Table 4 clearly shows that data augmentation is really helpful, and I think a fair comparison would apply data augmentation to the other methods as well.
- W3. I don't see a reason to run only three seeds on these experiments, as they are not that heavy. Also, there are not statistical tests performed, but rather the best on-average results are marked in bold.
- W4. While sensible, the theoretical analysis makes quite some assumptions. I am particularly concerned with how realistic is the first point of assumptions 2 ($\nabla_\theta^2 \log f(y|x,\theta)$ is independent of the class labels $y$). However, while I find the toy example really confusing (more on that later), Figure 3 is quite helpful.

**Questions/Comments:**
- Q1. Results on CCIFAR, CMNIST, and BAR on Vanilla and LfF are significantly different from those reported by LfF. Could the authors explain where do the differences come from?
- Q2. Are you sure the cross-entropy loss is right? I think it is missing terms. See, for example. [here](https://ml-cheatsheet.readthedocs.io/en/latest/loss_functions.html).
- Q3. What is "an official dataset (e.g. MNIST)" supposed to mean? What does a dataset "official"?
- Q4. Was data augmentation used in the appendix C7?

**Summary Of The Paper:**

Machine Learning models tend to take shortcuts when learning, taking advantage of dataset biases (or spurious correlations) within the data features, that make the task considerably easier to solve. However, these correlations are, indeed, nothing but spurious, and they may not be present during testing, making the model brittle and prone to errors. Existing approaches to solve this dataset bias problem focus either on leveraging human-labelled information, or on using attributes of the network such as their output, loss values, or feature space. This work, instead, focuses on the gradient of the samples with respect to the model parameters, and proposes a novel approach which: i) trains a biased model; ii) detects the biased samples based on their gradient; and iii) trains a new model with importance sampling to correct these biases. Theoretical and empirical results show the effectiveness of the proposed method.

**Summary Of The Review:**

I think this is an interesting paper that addresses an important problem. However, I feel that it is a result of a number of systematic steps starting from LfF, and it is not entirely clear whether all the steps are necesssary to obtain good results (specially the gradient norms, which have not been tested in isolation).

This, combined with a lack of clarity, and the need to clarify some experimental results, lean me towards rejection for now. I am sure a polished version of the current work will be a pretty good paper.

---

> ### Author Response · Authors · 2022-11-12
> **Response to Reviewer oD49 (3/3)**
>
> ---
> **Q9) About Non-probabilistic robust optimization**
>
> A9)  Thank you for the constructive comment. We agree that eq (6) can be interpreted as a non-probabilistic robust optimization, where the decision is pmf $h(x) \in \mathcal{H}$, and nature is the test marginal distribution $q(x)$.
> We have added the discussion with some related works [11,12,13] that solve the dataset bias problem as a robust optimization: “(6) is a min-max problem formula, a type of robust optimization. Similar problem formulations for solving the dataset bias problem can be found in [11,12,13]. However, they assume that the training data is divided into several groups, and the model minimizes the worst inference error of the reweighted group dataset. In contrast, the objective of (6) minimizes the worst-case test loss without explicit data groups where the test distribution can be arbitrary”.
>
> ---
> **Q10) Readability includes typos and incorrect grammars.**
>
> A10) Thank you for your constructive feedback on readability. First of all, we modified the current manuscript to improve the readability. Moreover, we had the entire manuscript using a professional editing service.
>
>
>
> [1] Learning from Failure: Training Debiased Classifier from Biased Classifier, NeurIPS 2020
>
> [2] Why resampling outperforms reweighting for correcting sampling bias with stochastic gradients, ICLR 2021
>
> [3] Generalized cross entropy loss for training deep neural networks with noisy labels, NeurIPS 2018
>
> [4] Learning De-biased Representations with Biased Representations ICML 2021
>
> [5] Learning Debiased Representation via Disentangled Feature Augmentation, NeurIPS 2021
>
> [6] Convergence Rates of Active Learning for Maximum Likelihood Estimation, NeurIPS 2015
>
> [7] Asymptotic Analysis of Objectives based on Fisher Information in Active Learning, JMLR 2017
>
> [8] A Probabilistic Active Learning Algorithm Based on Fisher Information Ratio, IEEE Transaction 2018
>
> [9] Gone Fishing: Neural Active Learning with Fisher Embeddings, NeurIPS 2021
>
> [10] ImageNet-trained CNNs are biased towards texture; increasing shape bias improves accuracy and robustness, ICLR 2019
>
> [11]  Invariant Risk Minimization, arXiv 2020
>
> [12]  Predict then Interpolate: A Simple Algorithm to Learn Stable Classifiers, ICML 2021
>
> [13] Rich Feature Construction for the Optimization-Generalization Dilemma, ICML 2022

---

> > ### Comment · Reviewer_oD49 · 2022-11-14
> > **Thank you for the answers**
> >
> > Dear authors, thanks a lot for the grealy detailed answer. Most of my concerns have been nicely resolved, especially Q1 in your answer, I think those new experiments show that using the gradient norm is indeed also benefitial. I will update my score.
> >
> > I still have to disagree with the authors in the statistical significance of the results. Three evaluations tell us almost nothing regarding the sensitivity of the algorithm, especially when there are standard deviations as big as those from Table 1. Assuming the evaluations are Gaussian, 95% of the data lie within 2 stds of the mean (68% within 1 std). In any case, running a statistical test (e.g. t-test) is quick, easy, and less handwavy than comparing intervals.
> >
> > With that said, I understand that resources are limited and that there are a lot of experiments, but I find the highlighting biased towards your method (which is unnecessary, the method performs great). Also, for this particular matter, I personally don't like using previous work as an argument to get away with running 3 seeds, as it is a _bad practice independently of what others do_. Not that long ago, researchers were running experiments on fine-tuned random seeds.
> >
> > Regarding fair comparisons, my main concern is that data augmentation clearly helps. In other methods without resampling, data augmentation will work mostly on dominant classes, which I feel can give more of an edge to your method. However, I understand that being able to effectively apply data augmentation is a benefit of resampling over re-weighting.
> >
> > Also, I have a few follow-up questions:
> > 1. When performing the ablation studies, did you keep data augmentation?
> > 2. Why not keeping the standard deviation of the new table in the appendix?

---

> > > ### Author Response · Authors · 2022-11-15
> > > **Answer to the reviewer's additional questions**
> > >
> > > Dear reviewer oD49,
> > >
> > > We would like to thank you for your constructive and insightful feedback. We address your further concerns and questions below.
> > >
> > > **Q1) What are the T-test results of PGD and other algorithms (e.g., Disen)?**
> > >
> > > A1) In order to alleviate your concern, we conducted the t-test through experiments with several seeds on CMNIST. In particular, we run PGD and Disen (the best performing algorithm among baselines) algorithms 25 times each to obtain the following statistics. Note that we use the same setting as that in Table 1 of the main manuscript. Based on the experimental results, we conducted t-test and obtained the following p-value. Note that our alternative hypothesis is “PGD outperforms Disen”; thus lower p-value means that PGD is less likely to be worse than Disen. Furthermore, we report max, min statistics of 25 trials for more information..
> > >
> > > |              | p-value | min Disen | max Disen | min PGD | max PGD |
> > > |--------------|:-------:|:---------:|:---------:|:-------:|:-------:|
> > > | $\rho = 0.5$ | 0.01572 |   92.27   |   95.35   |  94.27  |  97.18  |
> > > |  $\rho = 1$  | 0.01239 |   96.09   |   97.47   |  98.01  |  98.70  |
> > > |  $\rho = 5$  | 0.29370 |   97.52   |   98.60   |  98.47  |  98.80  |
> > >
> > >
> > > As a result of the p-values, the reliability of PGD to perform better than Disen is high, especially lower $\rho$ cases. This means that PGD performs well with high probability. We added this information at Appendix D.3.
> > >
> > >
> > > **Q2) Is the same augmentation used in other case-studies?**
> > >
> > > A2) Yes it is. To clarify this, we specified that “the same setting as the experiment in Table 1 was applied except for the special case we want to observe” at the beginning of Appendix C.
> > >
> > > **Q3) Standard deviation report in case-study Tables.**
> > >
> > > A3) We agree that additional standard deviations can provide better reliability. To improve the clarity, we added standard deviation for all study cases in Appendix C. Thank you for pointing out what we missed.

---

> > > > ### Comment · Reviewer_oD49 · 2022-11-15
> > > > **Thanks for the responses**
> > > >
> > > > Dear authors,
> > > > Thanks for the follow-up response and the time spent on running the statistical tests. My concerns are mostly resolved by now and I will udpate my recommendation accordingly.

---

> > > > > ### Author Response · Authors · 2022-11-16
> > > > > **Thank you for your feedbacks**
> > > > >
> > > > > We appreciate your positive assessment of our work. Thanks to you, we were able to improve the clarity of our deliverable manuscript.

---

> ### Author Response · Authors · 2022-11-12
> **Response to Reviewer oD49 (2/3)**
>
> ---
> **Q4) Concerns about no statistical performance analysis other than average performance**
>
> A4) We reported the standard deviation and the average value, and the overlapping standard deviation (Disen in BAR case) was also highlighted as an underline, to provide statistical analysis as well as average performance analysis. The observed standard deviations, the standard deviation of PGD did not overlap most of the other baselines, which can be interpreted as them being statistically less likely to show lower performance than other methods.
>
> ---
> **Q5) About viability of Assumption 2**
>
> A5) The label-independent Hessian assumption is frequently used in active learning research areas [6-9]. In many probabilistic models, including linear and logistic regression, indeed, the Hessian of the loss function does not depend on the label $y$, as discussed in Section 5 of [6]. We believe that our theoretical analysis based on this assumption is meaningful, at least for motivating PGD.
>
> ---
>
> **Q6) The missing notation description for the definition of cross-entropy loss in page 2**
>
> A6) The cross-entropy we used is a term with a one-hot encoded label. We should have included the information. If the label is one-hot, the cross-entropy term of the link that the reviewer suggested is equivalent to the term we wrote the following flows:
>
> *Notation*
> - $C$: total number of classes
> - $x_i, y_i$ : i-th input  image and its corresponding true class
> - $Y_i=${$Y_{i,c}$} : $C$ dimension one-hot encoded probability vector for i-th data, impose probability 1 to true class $y_i$. i.e., $Y_{i,y_i}=1$
> - $f(c| x_i, \theta)$: Model softmax output for $c$-th class
>
> By using the above notations, the cross- entropy loss in the suggested link can be written as follows:
>
> $L_{\text{CE}}(x_i,y_i;\theta)=-\sum_{c=1}^C Y_{i,c} \times \log f(c|x_i, \theta)$
>
>
> Due to $Y_{i,c} = 0$ when $c \neq y_i $ and $Y_{i,y_i} = 1$ , the above definition can be re-written as follows:
>
>  $L_{CE}(x_i, y_i; \theta) = - \log f(y_{i} | x_i, \theta)$,
>
> which is our definition in page 2.
> We revised this part for a clearer description, added to revision, “when the label is one-hot encoded” as described in Section 2. Thank you very much for pointing this out.
>
> ---
> **Q7) Sources of datasets**
>
> A7) The source of each dataset is as follows:
> - CMNIST: we randomly generated CMNIST. The protocols are the same with the previous papers [1,5]. The only difference is the colors which are uniform randomly selected.
> - CCIFAR: We downloaded the CCIFAR dataset from the officially offered link from the Disen [5] paper. This is generated by following the protocol presented in LfF [1].
> - BAR, BFFHQ: We also downloaded both datasets from the officially offered link from LfF [1] and Disen [5], respectively.
>
> We misused the term “official data” and caused confusion. Regarding this, we revised the manuscript more clearly.
>
>
>
> ---
> **Q8) Why is the performance of LfF different from the value reported in the LfF paper?**
>
> A8) This is due to the different architecture (CMNIST, CCIFAR), dataset (CCIFAR), additional augmentation (CMNIST, CCIFAR, BAR), and the different hyperparameters (BAR). We justify our experimental setting as follows:
> - CMNIST: We utilized CNN rather than MLP, because it is argued that CNN is vulnerable to the texture (e.g., color) bias [10]. In addition, while LfF did not utilize augmentations, we apply augmentations, (i.e., color jitter, resize crop, and rotation). Note that we applied the identical augmentations to all methods for fair comparison.
> - CCIFAR: We followed the setting of Disen [5], a more recent setting-experimenting CCIFAR. For example, we downloaded the dataset from the official link, which is offered by the authors of [5], and used ResNet-18 instead of ResNet-20. As mentioned by the authors of [5], the CCIFAR dataset is constructed by following the protocol of LfF except for the type of corruptions. We believe that our implementation of LfF is not wrong based on the performance is similar to the reported values in [5] (e.g., 28.83% in our paper and 28.57% when $\rho=0.5\%$).
> - BAR: Two different points, hyperparameter and augmentation, generate the performance difference. First of all, while LfF used random resize crop, we used color jitter, resize crop and rotation for all baselines. Furthermore, we change the hyperparameter that shows better performance than the reported value, and compare by using the searched parameters.
>
> Due to the recent paper settings as written in the above, and additional modules that can help improve the performance of LfF, some improvements are reported; we believe that this comparison is fair because above modules are all methods-friendly, including LfF.

---

> ### Author Response · Authors · 2022-11-12
> **Response to Reviewer oD49 (1/3)**
>
> We are very grateful for your constructive comments. We have re-written your comments as we understand them and provide corresponding answers. Please let us know if there are any misunderstandings or if you have additional questions.
>
> ---
> **Q1) Originality of PGD compared to LfF [1]**
>
> A1) I would like to emphasize that PGD has the following originalities compared to LfF.
> 1. **Two stage approach**: While LfF performs debiasing using statistics from the evolving biased model as training goes on, PGD enables stable training by utilizing statistics, i.e., per-sample gradient, obtained from the sufficiently converged biased model. From the stable training, PGD can improve debiasing performance. As described in Appendix C.6 of the first submission, it can be confirmed by comparing two-stage and single-stage performance. Note that this section has been moved to Appendix C.2 of the revision.
> 2. **Resampling based approach**: While LfF is a reweighting method with different learning weights for each sample, PGD is a resampling method that construct the batch by adjusting the sampling distribution, and there is a mathematical analysis argument that the resampling method is more stable than reweighting [2]. This argument also worked under our setting, as described in Appendix C.7 of the first submission. Note that this section has been moved to Appendix C.3 of the revision.
> 3. **Gradient based approach**: PGD uses information from the parameter space wider than the loss space, which is utilized in LfF, when training the debiased model. Therefore, we argue that we can expect PGD to have better performance than LfF. To address the reviewer’s concern, we set-up additional experiments to verify the superiority of gradient and report the results of the experiment as follows:
> - PGD with loss: This is a method of sampling proportional to the per-sample loss of the trained biased model, instead of per-sample gradient norm in PGD.
> - LfF with per-sample gradient: In LfF, the relative difficulty values are with the per-sample gradient norm. In other words, per-sample gradient norm is utilized to give weight to each sample for training the debiased model.
>
> | CMNIST              | \rho=0.5%        | \rho=1%          | \rho=5%          |
> |---------------------|------------------|------------------|------------------|
> | LfF                 | $91.35 (\pm 1.35)$ | $96.88 (\pm 0.20)$ | $98.18 (\pm 0.05)$ |
> | PGD + loss  |$30.63 (\pm 2.23)$ | $34.04 (\pm 3.00)$  | $78.48 (\pm 1.41)$ |
> | LfF + gradient norm | $93.29 (\pm 0.39)$ |$97.55 (\pm 0.24)$ | $98.37 (\pm 0.20)$ |
> | PGD                 | $96.88 (\pm 0.28)$ | $98.35 (\pm 0.12)$ | $98.62 (\pm 0.14)$ |
>
> As shown in the above table, the best method is PGD, and the second best is the case where per-sample gradient norm is utilized in LfF. This means that the per-sample gradient norm is better than the relative difficulty of LfF. Moreover, based on the case of where loss values are used to compute the sampling probability, loss value is not appropriate for the resampling method. We added the results of the above table to Appendix C.4 of the revision.
>
> To clarify, the thing we are motivated by LfF [1] is the usage of GCE [3] to intensify the bias for training the biased model. For a much clearer description about GCE, we added to revision, “the biased model, $f_b$, is trained on the generalized cross-entropy (GCE) loss $L_{\text{GCE}}$ to amplify the bias of the biased model, motivated by [1].” at the Section 3.
>
>
> ---
> **Q2) Data augmentation setting on other baselines.**
>
> A2) For fair comparison, we used the same augmentation operations of PGD for all baselines. In particular, we used color jitter and random rotation for CMNIST, CCIFAR, MBMNIST, BAR, and BFFHQ cases.  In the analysis part of Appendix C.7 (now moved to Appendix C.3 in the revised manuscript), all settings, such as data augmentation, except for the reweighting related part, were used in the same way as in the resampling case. For a clearer explanation of the settings, we wrote at the beginning of Appendix C that all experimental settings for each analysis are the same with an exception for each analysis.
> Therefore, we believe that our experimental results are fair compared by giving augmentations to all baselines.
>
> ---
> **Q3) The reason why we ran three independent seeds.**
>
> A3) We have tested algorithms in various benchmarks. For example, accuracy comparison on 7 datasets in Table 1-3, ablation studies in Table 4, gradient-norm type analysis in Figure 2, and further analysis in Appendix C of the first submission are conducted. Therefore, reporting statistics more than three seeds to verify the sensitivity of the algorithm was already very heavy for our limited computing resources. Moreover, we would like to recall that many existing studies [1,4,5] also have verified sensitivity statistics through three trials.

---

### Official Review · Reviewer_jyHT · 2022-11-02

**Confidence:** 3
**Correctness:** 3
**Technical Novelty And Significance:** 2
**Empirical Novelty And Significance:** 2
**Recommendation:** 8

**Clarity, Quality, Novelty And Reproducibility:**

The paper is well written. The novelty lies in the usage of gradient norms. It has sufficient details in the paper and also provided code for reproducibility.


**Strength And Weaknesses:**

Strengths:

[S1] The approach is simple and logical. It leverages the fact that rarer samples have higher gradient norms. This seems to be the first paper to use gradient norm instead of the commonly used output space-based measures for re-sampling/re-weighting.

[S2] Mathematically, the paper interprets the debiasing problem as that of min-max (minimizing loss) of maximally difficult samples, which can be relaxed to minimize the trace of inverse Fisher Info. They show how the gradient norm minimizes this theoretically for 1D and empirically for higher dimensions. This provides a strong justification to the choice of gradient norms.

[S4] The results on image classification datasets used by previous debiasing methods and Civil Comments (from WILDs) show gains over existing methods. However, I have a concern in [W1]

Weaknesses:

[W1] Sec 4.2 mentions 8 baselines, but not all comparison methods are run on all the datasets. For example, EIIL seems to be reported only for CivilComments and BPA only for CelebA. I did not find a good justification for this in the paper; it seems like all the methods can potentially be run on all the datasets.



**Summary Of The Paper:**

The paper debiases models by resampling with probabilities proportional to gradient norms. It uses the generalized cross-entropy loss from LfF to train a biased model and debiases the main model by sampling probabilities from gradient norms. It leverages the observation that rarer samples tend to have higher gradient norms, so their resampling scheme assigns higher weights to rarer samples.

**Summary Of The Review:**

Overall, I think the approach is simple and has practical applications -- from the results it seems to be the SoTA amongst all the implicit de-biasing methods.
However, the paper does not report all their comparison methods on all the datasets, which is really strange.
I am willing to raise the score if the authors properly justify why that is the case.

---

> ### Author Response · Authors · 2022-11-12
> **Response to Reviewer jyHT**
>
> We are very grateful for your constructive comments. We have re-written your comments as we understand them and provide corresponding answers. Please let us know if there are any misunderstandings or if you have additional questions.
>
> **Q1) The reason why there are some blanks in Table 3.**
>
> A1) We followed the setting of BPA [1] for CelebA and the setting of CNC [2] for CivilComments-WILDS. [1] and [2] were the most recent works at the time of submission. Although CNC also mentioned CelebA's performance, the model implemented in BPA and CNC are different. So, we cannot directly compare BPA and CNC.  (BPA: implemented in ResNet18 settings; CNC: implemented in ResNet 50 settings) To our best knowledge, BPA is the only algorithm implemented in CelebA of BPA setting. Since BPA setting has few baselines, we replace the results of CelebA by the setting implemented in CNC. Table 3 is updated as a result of table below, and CelebA results from the existing BPA setting are moved to Appendix E.1.
>
> ||Vanilla|LfF|EIIL|JTT|CNC|Ours|
> |:-:|:-:|:-:|:-:|:-:|:-:|:--:|
> |CelebA Avg.|94.9|85.1|85.7|88.1|88.9|88.6|
> |CelebA Worst|47.7|77.2|81.7|81.5|**88.8**|**88.8**|
> |CivilComments Avg.|92.1|92.5|90.5|91.1|81.7|92.1|
> |CivilComments Worst|58.6|58.8|67.0|69.3|68.9|**70.6**|
>
> As shown in the above table, PGD shows CNC-like performance in CelebA. However, we argue that the training overhead of PGD may be more negligible than CNC because CNC utilizes a contrastive learning framework. Moreover, PGD shows the best performance against all baselines in CivilComments-WILDS.
>
> [1] Unsupervised learning of debiased representations with pseudo-attributes, CVPR 2022
>
> [2] Correct-N-contrast: A contrastive approach for improving robustness to spurious correlations, ICML 2022

---

> > ### Comment · Reviewer_jyHT · 2022-11-28
> > **Response**
> >
> > Thank you authors for the response.
> > My original remark corresponded to this statement in the original submission:
> > "Our baselines comprise eight methods on the various tasks: ... ".
> >
> > I was confused since George, EIIL, CNC were not reported in Table 1. Perhaps I missed it, but is there a section discussing which algorithms are not run on certain datasets and the reasons?

---

> > > ### Author Response · Authors · 2022-11-29
> > > **Answer to the reviewer's additional question**
> > >
> > > Dear reviewer jyHT,
> > >
> > > We would like to thank you for your clarified question. We address your further question below.
> > >
> > > **Q) Reason why some algorithms (EIIL, CNC, George) were only compared in large scale datasets (e.g., CelebA, CivilComments).**
> > >
> > > A) We aimed to broaden comparisons with various methodologies on different datasets with limited computational resources. However, because there were fewer tasks that were shared among different baselines, a performance analysis was done on as many tasks as possible (seven tasks) to compare the proposed method with various methods. As a result, some of the methods were conducted only on the large scale dataset.
> > >
> > > To address the reviewer’s concern, we additionally report the results of the CMNIST performance that is partially compared in CNC as follows. Note that the protocol for generating CMNIST and training recipe differs from the our previous setting (e.g., reduce the number of classes to five, use LeNet-5, learning rate 1e-3, batch size 32, weight decay 5e-4, momentum 0.9, training 5 epochs). We conducted an experiment by generating datasets with the code provided by the author of CNC.
> > >
> > > |            | Unbiased test Accuracy | Worst case Acc 		|
> > > |------------|------------------------|------------------------|
> > > | Vanilla*    | 20.1 $\pm$ 0.2         | 0.0 $\pm$ 0.0  		|
> > > | LfF*        | 25.0 $\pm$ 0.5         | 0.0 $\pm$ 0.0  		|
> > > | George*     | 89.5 $\pm$  0.3        | 76.4 $\pm$ 2.3 		|
> > > | EIIL*       | 90.7 $\pm$  0.9        | 72.8 $\pm$ 6.8 		|
> > > | JTT*        | 90.2 $\pm$  0.8        | 74.5 $\pm$ 2.4 		|
> > > | CNC*        | 90.9 $\pm$  0.6        | 77.4 $\pm$ 3.0 		|
> > > | PGD (Ours) | **93.1$\pm$0.3**       | **82.5 $\pm$ 2.9** 	|
> > >
> > > $*$ mark denotes the reported value in CNC paper. As stated in the above table, PGD is performing well on a similar dataset (CMNIST with 5 digits) provided by CNC. In order to clarify this, we will add the above results in the final manuscript. Thank you.
> > >
> > > Sincerely,
> > >
> > > Authors.

---

> > > > ### Comment · Reviewer_jyHT · 2022-11-30
> > > > **Post-Rebuttal Response**
> > > >
> > > > Thank you authors for addressing this concern. I have increased the score to 8, since the issue just needed some clarification.

---

> > > > > ### Author Response · Authors · 2022-11-30
> > > > > **Thank you for your positive assessment**
> > > > >
> > > > > We appreciate you taking the time to read our response and manuscript.
> > > > >
> > > > > We will keep working on developing more formal technical statement.
> > > > >
> > > > > Sincerely,
> > > > >
> > > > > Authors.

---

### Author Response · Authors · 2022-11-12
**General Response**

Dear reviewers,

We wish to thank all reviewers (**@jyHT, @oD49, @Lo2Z**) sincerely for carefully reviewing our paper and giving insightful and constructive comments. We are genuinely encouraged to propose a simple (**@jyHT, @oD49, @Lo2Z**) and novel perspective (**@jyHT, @Lo2Z**) approach of solving the dataset bias problem with mathematical evidence (**@jyHT, oD49**). Furthermore, we are pleased that (**@jyHT, @Lo2Z**) feel our paper is clearly written and easy to understand.

We addressed the concerns from reviewers in each reply, and modified our manuscript reflecting this as follows:

---
- Fill in the blanks in Table 3, for **@jyHT** (Section 4)
- Performance verification of per-sample gradient, for **@oD49** (Appendix C. 4)
- Explanation relevance to robust optimization, for **@oD49** (Section 5)
- Clearer description about experimental settings, for **@oD49** (Section 4, Appendix A, Appendix C)
- More description of the definition of CE loss, for **@oD49** (Section 2)
- Clarify the use of GCE, which is motivated by LfF, for **@oD49** (Section 3)
- Readability improvement, correct typos, and much clearer description
---

The revisions made are marked with $\color{purple}{purple}$ colors in the revised paper.

Thank you for your time,

Sincerely,

Authors

---

### Decision · Program_Chairs · 2023-01-20

**Decision:**

Accept: poster

**Justification For Why Not Higher Score:**

It a strong poster paper.

**Justification For Why Not Lower Score:**

The reviewers have raised their score from rejection to acceptance after rebuttal. All reviewers recommend acceptance.

**Metareview: Summary, Strengths And Weaknesses:**

This paper proposes a debiasing framework using per-sample gradient to reweigh data points in proportion to the gradient norm. It departs from the trend of label-based resampling/reweighing and leverages the fact that rarer samples have higher gradient norms in the biased model. Theoretical motivation based on Fisher information has been developed.  Extensive empirical evaluations and strong rebuttal convinced all the reviewers. The reviewers and AC found the contributions sensible and recommend acceptance. We urge the authors to include the clarifications of the rebuttal into the manuscript, highlighting the response to the main concerns in the main paper, if possible. Congratulations to the authors!

**Note From Pc:**

if the above contains the word "oral" or "spotlight" please see: "oral" presentation means -> notable-top-5% and "spotlight" means -> notable-top-25%. As stated in our emails, we are disassociating presentation type from AC recommendations